# RNA localization mechanisms transcend cell morphology

Raeann Goering[1,2], Ankita Arora[1], Megan C Pockalny[1], J Matthew Taliaferro[1,2]*

[1]Department of Biochemistry and Molecular Genetics, University of Colorado Anschutz Medical Campus, Aurora, United States; [2]RNA Bioscience Initiative, University of Colorado Anschutz Medical Campus, Aurora, United States

**Abstract** RNA molecules are localized to specific subcellular regions through interactions between RNA regulatory elements and RNA binding proteins (RBPs). Generally, our knowledge of the mechanistic details behind the localization of a given RNA is restricted to a particular cell type. Here, we show that RNA/RBP interactions that regulate RNA localization in one cell type predictably regulate localization in other cell types with vastly different morphologies. To determine transcriptome-wide RNA spatial distributions across the apicobasal axis of human intestinal epithelial cells, we used our recently developed RNA proximity labeling technique, Halo-seq. We found that mRNAs encoding ribosomal proteins (RP mRNAs) were strongly localized to the basal pole of these cells. Using reporter transcripts and single-molecule RNA FISH, we found that pyrimidine-rich motifs in the 5′ UTRs of RP mRNAs were sufficient to drive basal RNA localization. Interestingly, the same motifs were also sufficient to drive RNA localization to the neurites of mouse neuronal cells. In both cell types, the regulatory activity of this motif was dependent on it being in the 5′ UTR of the transcript, was abolished upon perturbation of the RNA-binding protein LARP1, and was reduced upon inhibition of kinesin-1. To extend these findings, we compared subcellular RNAseq data from neuronal and epithelial cells. We found that the basal compartment of epithelial cells and the projections of neuronal cells were enriched for highly similar sets of RNAs, indicating that broadly similar mechanisms may be transporting RNAs to these morphologically distinct locations. These findings identify the first RNA element known to regulate RNA localization across the apicobasal axis of epithelial cells, establish LARP1 as an RNA localization regulator, and demonstrate that RNA localization mechanisms cut across cell morphologies.

*For correspondence:
matthew.taliaferro@cuanschutz.
edu

Competing interest: The authors declare that no competing interests exist.

## Editor's evaluation

In this valuable work, the authors used a combination of RNA-seq-based approaches and reporter mRNAs coupled to RNA imaging, to provide solid evidence that mRNAs with specific elements (TOP, GA, and NET1 3'UTR) localize asymmetrically across species and cell types and that this is likely mediated by conserved RBPs and via direct transport mechanisms preferentially involving kinesin motors. The work will be of interest to the cell biology community focused on mRNA localization.

## Introduction

The post-transcriptional regulation of RNA allows for the fine tuning of the expression of genetic information. These post-transcriptional processes, including alternative splicing, translational regulation, RNA decay, and subcellular RNA localization, are often regulated by RNA binding proteins (RBPs) that exert their function upon their RNA targets by recognizing and binding specific sequence motifs. Because the binding of RNA by RBPs is largely governed by specific sequence-based interactions,

**eLife digest** The information required to build a specific protein is encoded into molecules of RNA which are often trafficked to precise locations in a cell. These journeys require a complex molecular machinery to be assembled and set in motion so that the RNA can be transported along dynamic 'roads' called microtubules. The details of this mechanism are known only for a handful of RNAs in a few cell types; for example, scientists have uncovered the signals presiding over the shuttling of certain RNAs to the axon, the long and thin projection that a neuron uses to communicate. Yet these RNAs are also present in cells that lack axons. Whether the molecular processes which preside over RNA movement apply across cell types has so far remained unclear.

To investigate this question, Goering et al. tracked the location of RNA molecules in two types of polarized mouse cells: neurons which feature an axon, and 'epithelial' cells which line the intestine. The experiments revealed that the signals sending RNAs to the axons also directed the molecules towards the bottom pole of epithelial cells. In both cases, the RNAs travelled towards the extremity of the growing, "plus" end of the microtubules.

Overall, this work suggests that RNA transport mechanisms should not be thought of as leading to a particular location in the cell; instead, they may be following more generalisable instructions. This knowledge could allow scientists to predict where a particular RNA will be sent across cell types based on data from one cell population. It could also aid the development of synthetic RNAs that target specific parts of the cell, offering greater control over their actions.

RBPs often bind similar RNA targets across cell types. Therefore, RBPs often exert their regulatory function upon the same RNA targets, regardless of cellular context (*Matia-González et al., 2015*; *Van Nostrand et al., 2020*).

Many RBPs are ubiquitously expressed across tissues and generally elicit the same post-transcriptional regulatory behaviors on their target RNAs in many cell types (*Gerstberger et al., 2014*). The subcellular localization of an RNA, though, is at least conceptually tightly linked to cell morphology. If the interaction between an RBP and an RNA sequence in neurons results in a transcript's localization to axons, where will the same transcript localize in a cell type that doesn't have axons yet nevertheless expresses the same regulatory RBP? (*Figure 1A*) Can we define spatial equivalents for RNA destinations across cells with vastly different morphologies?

For a given RNA, much of what is known about the regulation of its localization, especially in terms of sequence motifs and the RBPs that bind them, has been elucidated in a single cell type. Many examples of the molecular rules that govern RNA transport to neuronal projections (*Arora et al., 2022a*; *Goering et al., 2020*; *Von Kügelgen et al., 2021*; *Mikl et al., 2022*; *Zappulo et al., 2017*) or across the posterior/anterior axis of *Drosophila* embryos *Ephrussi and Lehmann, 1992*; *Hachet and Ephrussi, 2004*; *Jambor et al., 2015*; *Lécuyer et al., 2007*; *Lipshitz and Smibert, 2000* have been defined. However, how these RBPs and RNA elements would regulate RNA localization in cell types or morphologies outside of those that they were discovered in is generally unknown.

RNA localization is widely studied in neuronal models as their long projections are amenable to both microscopy and subcellular fractionation. Hundreds of RNAs are actively transported into neuronal projections (*Cajigas et al., 2012*; *Gumy et al., 2011*; *Taliaferro et al., 2016*; *Zappulo et al., 2017*; *Zivraj et al., 2010*). Many of these RNAs are not neuron specific and are expressed widely throughout many other tissues (*Von Kügelgen and Chekulaeva, 2020*; *Zappulo et al., 2017*). Again, whether these RNA are localized in other cell types is largely unknown.

Intestinal enterocytes line the intestine as a single-cell monolayer. Enterocytes are polarized across their apicobasal axis with specialized regions that perform specific functions (*Klunder et al., 2017*). The apical regions of enterocytes are responsible for nutrient absorption from the gut. whereas the basal regions export processed nutrients to the lymph and bloodstream (*Chen et al., 2018*; *Goodman, 2010*). Many proteins and RNAs are asymmetrically localized across the apicobasal axis to aid in these spatially distinct functions, but little is known about the RNA elements and RBPs that regulate a transcript's location within epithelial cells (*Moor et al., 2017*; *Ryder and Lerit, 2018*).

Using transcriptome-wide assays, we reveal that the neurites of neurons and the basal pole of epithelial cells are enriched for similar RNAs. We further find that specific RNA regulatory elements

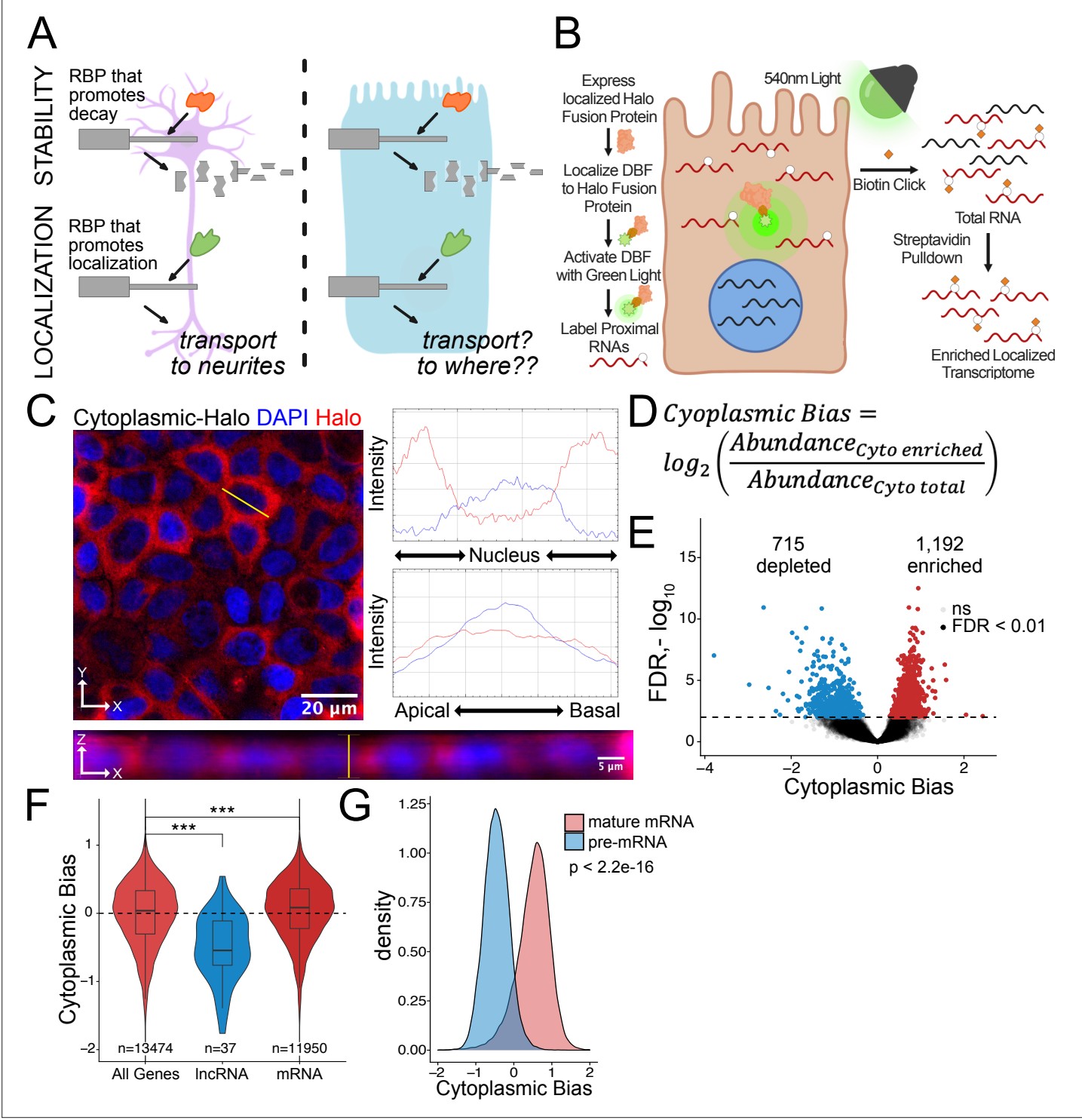

**Figure 1.** Halo-seq enriches cytoplasmic localized RNA molecules in C2bbe1 monolayers. (**A**) RNA regulatory processes are generally driven through the interaction of an RBP and an RNA sequence element. Because RBPs are often widely expressed, many of these processes (exemplified here by RNA decay) operate across cellular contexts. The orange RBP binds a particular sequence within a transcript to promote its degradation in both cell types. However, because RNA localization is intimately linked with cell morphology, if an RBP/RNA interaction promotes RNA localization in one cell type, how the same interaction would affect RNA localization in another cell type is unclear. (**B**) Overview of the Halo-seq method. A HaloTag is genetically fused to a protein with specific localization. DBF-conjugated Halo ligands are specifically recruited to the Halo fusion proteins due to their high affinity and covalent binding to HaloTag protein domains. When exposed to green light, DBF produces oxygen radicals that, in combination with propargylamine, alkynylate DBF-proximal RNAs. 'Click' chemistry covalently attaches biotin moieties to alkynylated RNAs in vitro, facilitating their enrichment via

*Figure 1 continued on next page*

Figure 1 continued

streptavidin pulldown, thereby enriching localized RNAs for high-throughput sequencing. (**C**) HaloTag protein fusions to P65 are localized to the cytoplasm. HaloTag fusions are visualized in fixed cells through the addition of a Halo ligand fluorophore shown in red. DAPI stain marks the nuclei in blue. Profile lines across the XY image demonstrate exclusion of Halo signal from the nucleus and in the XZ-projected image demonstrate no apical or basal bias. Images are an average projection through the XZ axis of approximately 50 cells. (**D**) Equation for Cytoplasmic Bias. Cytoplasmic RNA localization is calculated with a metric termed Cytoplasmic Bias as the $\log_2$ of a transcript's abundance in the streptavidin enriched fraction divided by its abundance in the input total RNA. (**E**) Genes with differing Cytoplasmic Bias following Halo-seq RNA labeling using the cytoplasmic-Halo fusion. FDR calculated by DESeq2 (**F**) Cytoplasmic Bias for defined classes of RNAs. p-Values were calculated using a Wilcoxon rank-sum test. (**G**) Cytoplasmic Bias for unspliced, intron-containing pre-mRNAs and spliced mature mRNAs. Cytoplasmic Bias was calculated for the spliced and unspliced isoform of each transcript. p-Values were calculated using a Wilcoxen rank-sum test. ns (not significant) represents $p > 0.05$, * $p < 0.05$, ** $p < 0.01$, *** $p < 0.001$ and **** represents $p < 0.0001$.

The online version of this article includes the following source data and figure supplement(s) for figure 1:

**Figure supplement 1.** Schematic of Lox cassette knock-in to the AAVS1 safe harbor locus.

**Figure supplement 2.** PCR amplification of knock-in junctions.

**Figure supplement 2—source data 1.** 1% Agarose gel of PCR products using primers flanking the AAVS1 insertion site for the LoxP cassette.

**Figure supplement 3.** Schematic of cre-mediated cassette switching.

**Figure supplement 4.** SDS-PAGE gel of doxycycline-induced cytoplasmic-Halo construct (95 kDa) visualized with Halo ligand fluorophore in red (JF646).

**Figure supplement 4—source data 1.** GoPAGE Bis-Tris Precast Gel, 4–12% with Halo Ligand JF646 visualizing Halo tagged proteins in doxycycline induced cell lysates.

**Figure supplement 4—source data 2.** GoPAGE Bis-Tris Precast Gel, 4–12% with coomassie visualizing total protein extracted from the halo tag expressing cell lysates.

**Figure supplement 5.** Fusing Cy3-Azide fluorophores to alkynylated biomolecules using Click chemistry allows for visualization of labeled molecules in fixed cells.

**Figure supplement 6.** In vitro biotinylation of alkynyklated RNA labeled from cytoplasmic-Halo expressing cells is dependent on DBF addition as visualized by streptavidin-HRP on an RNA dot blot.

**Figure supplement 6—source data 1.** RNA is spotted and crosslinked onto an Amersham Hybond-N +western blotting membrane.

**Figure supplement 6—source data 2.** RNA is spotted and crosslinked onto an Amersham Hybond-N +western blotting membrane.

**Figure supplement 7.** Percent enriched RNA after streptavidin pull-down of cytoplasmic-Halo-labeled RNAs.

**Figure supplement 8.** Principal component analysis of gene expression values from Halo-seq in cytoplasmic-Halo cells.

and RBPs that regulate RNA localization in one cell type can predictably do so in other cell types with very different morphologies as well as in different species. These results hint at conserved mechanisms of RNA localization and potentially a predictable, underlying code for RNA localization compatible with vastly different subcellular locations and cellular morphologies.

# Results
## Halo-seq allows for isolation of spatially distinct RNA populations from C2bbe1 monolayers

C2bbe1 cells, a subclone of the human CaCo-2 cell line, recapitulate human intestinal enterocytes by spontaneously creating polarized monolayers with differentiated brush-borders in culture (*Peterson and Mooseker, 1992*). For easy genetic manipulation, we created a C2bbe1 cell line with a blasticidin resistance gene surrounded by *loxP* recombination sites in the AAVS1 safe harbor locus. This system allows for the controlled insertion of doxycycline-inducible transgenes using *cre* recombinase (*Khandelia et al., 2011*). This knock-in cell line was created with CRISPR/Cas9, an sgRNA targeting the AAVS1 locus, and a homologous repair donor plasmid containing the *loxP* cassette (*Figure 1—figure supplement 1*). Following selection with blasticidin, integration of the cassette was confirmed by PCR (*Figure 1—figure supplement 2*). This cell line is amenable to quick and efficient *cre*-mediated cassette switching allowing for stable doxycycline-inducible expression of any RNA or protein of interest (*Figure 1—figure supplement 3*).

We used our recently developed proximity labeling technique Halo-seq (*Engel et al., 2021*; *Lo et al., 2022*) to assay RNA localization across the apicobasal axis of C2bbe1 monolayers (*Figure 1B*). Halo-seq relies on the expression of a Halo-tagged protein with specific localization to a subcellular

compartment of interest. HaloTags are small protein domains that covalently bind to small molecules known as Halo ligands (*Los et al., 2008*). Through conjugation to a Halo ligand backbone, the photo-activatable singlet oxygen generator dibromofluorescein (DBF) can be specifically recruited to the Halo-tagged protein and therefore share the same subcellular distribution as the fusion protein.

When exposed to green light, DBF generates oxygen radicals that are restricted to a radius of approximately 100 nm around the Halo fusion protein (*Jin et al., 1995*; *Li et al., 2018*). RNA bases within this cloud are oxidized by the radicals, making them susceptible to nucleophilic attack by prop-argylamine, a cell permeable alkyne (*Li et al., 2018*). Halo-fusion-proximal RNAs are therefore selectively alkynylated in situ.

Following total RNA isolation, alkynylated transcripts are biotinylated using Cu(I)-catalyzed azide-alkyne cycloaddition (CuACC) 'Click' chemistry with biotin azide (*Hein et al., 2008*). Halo-proximal transcripts are therefore selectively biotinylated by this reaction allowing for purification of localized RNAs from total RNA via streptavidin pulldowns. By comparing the abundances of transcripts in total RNA and streptavidin-purified RNA samples using high-throughput sequencing, transcripts that were Halo-proximal can be identified.

To first assay the efficiency with which Halo-seq can enrich localized RNAs in C2bbe1 monolayers, we targeted the cytoplasmic compartment for analysis with Halo-seq. Because differences between the cytoplasmic and nuclear transcriptomes are well characterized (*Engel et al., 2021*; *Zaghlool et al., 2021*), we reasoned that this would provide a good benchmark of the technique in C2bbe1 monolayers. We fused a HaloTag domain to a cytoplasmically restricted NF-kappa B subunit, P65. This doxycycline-inducible transgene was integrated into the genome of C2bbe1 cells using cre/*loxP* recombination. The doxycycline-inducible expression and size of the cytoplasmic-Halo fusion were confirmed by gel electrophoresis with fluorescent Halo ligands (*Figure 1—figure supplement 4*). We also confirmed the cytoplasmic localization of the Halo fusion protein in situ using fluorescent Halo ligands (*Figure 1C*). We then visualized the site-specific alkynylation of biomolecules following DBF-mediated oxidation by performing azide-alkyne cycloaddition in situ with Cy3-conjugated azides. The presence of Cy3 labeled molecules required the addition of DBF, indicating that the observed alkynylation was due to the Halo-seq procedure (*Figure 1—figure supplement 5*). These experiments revealed that both the cytoplasmic-Halo fusion and the molecules it labeled were restricted to the cytoplasm, indicating that the DBF-mediated labeling was spatially specific.

To determine if we could biotinylate alkynated RNA, we performed Halo-seq in differentiated C2bbe1 monolayers expressing the cytoplasmic-Halo. Following in-cell alkynylation and in vitro Click-mediated biotinylation, we visualized biotinylated RNA using RNA dot blots with streptavidin-HRP. We observed strong RNA biotinylation signal that was dependent upon the addition of DBF to cells (*Figure 1—figure supplement 6*). We then incubated biotinylated RNA samples from DBF-treated and untreated cells with streptavidin-coated beads. For three DBF-treated samples, we recovered an average of 1.6% of the total RNA sample from the streptavidin-coated beads. In contrast, we only recovered an average of 0.05% of the total RNA sample from DBF-untreated samples, further illustrating that RNA biotinylation requires DBF treatment (*Figure 1—figure supplement 7*).

We then created rRNA-depleted RNAseq libraries from three streptavidin input and enriched RNA samples from DBF-treated cells. Following high-throughput sequencing, we quantified transcript and gene abundances in each sample using salmon and tximport (*Patro et al., 2017*; *Soneson et al., 2015*). As expected, samples separated primarily by fraction (input vs. streptavidin pulldown) in a principal component analysis (PCA) plot of transcript expression profiles (*Figure 1—figure supplement 8*).

To quantify cytoplasmic RNA localization, we created a metric called cytoplasmic bias (CB), which we defined as the $\log_2$ of a transcript's abundance in the streptavidin-pulldown samples compared to its abundance in the input RNA samples (*Figure 1D*). We used DESeq2 to identify genes that were significantly enriched in streptavidin-pulldown samples compared to input samples (*Love et al., 2014*). These represent genes whose transcripts were enriched in the cytoplasm. Using this approach, 1192 genes were identified as significantly cytoplasmically enriched (FDR < 0.01) while 715 genes were significantly depleted and therefore likely enriched in the nucleus (*Figure 1E*, *Supplementary file 1*).

We then looked at CB values for lncRNAs specifically. As a class, these transcripts are known to be enriched in the nucleus (*Lubelsky and Ulitsky, 2018*; *Shukla et al., 2018*), and therefore should be

depleted from the streptavidin-pulldown samples. As expected, lncRNA transcripts had significantly lower CB values, indicating that Halo-seq was faithfully reporting on RNA localization in C2bbe1 cells (*Figure 1F*). Additionally, we used a custom salmon index to simultaneously quantify intron-containing and fully spliced versions of every transcript. Spliced transcripts are exported to the cytoplasm while intron-containing transcripts are often restricted to the nucleus (*Luo and Reed, 1999*). Further validating Halo-seq, we observed that intron-containing pre-mRNA transcripts generally had negative CB values, indicating their nuclear enrichment. Conversely, intron-lacking, mature transcripts of the same genes had positive CB values, indicating their cytoplasmic enrichment (*Figure 1G*). These results provided confidence that Halo-seq identifies localized RNAs in C2bbe1 monolayers.

## Halo-seq identifies RNAs differentially localized across the apicobasal axis of C2bbe1 monolayers

To identify RNAs localized across the apicobasal axis of enterocytes, we first needed to ensure that we could establish polarized monolayers of C2bbe1 cells. We plated C2bbe1 cells on porous transwell inserts at high confluency and allowed spontaneous differentiation to occur for 7 days (*Masuda et al., 2011*; *Peterson and Mooseker, 1992*). This produced highly polarized monolayers with the expected spatially distinct localization of the polarity protein markers Ezrin and the sodium-potassium pump across the apicobasal axis, suggesting mature enterocyte-like organization (*Figure 2—figure supplement 1*).

To interrogate apicobasal RNA localization using Halo-seq, we created HaloTag fusion proteins that specifically localized to either the apical or basal poles of differentiated cells. In intestinal epithelial cells, podocalyxin-like protein 1 (PODXL) is apically localized while a subunit of the sodium-potassium pump (ATP1A1) is basally localized (*Deborde et al., 2008*; *Meder et al., 2005*). Using these membrane proteins, we created transgenes where HaloTags were fused to their N-termini such that the HaloTag domain was inside the cell. These doxycycline-inducible transgenes were integrated into the C2bbe1 genome using the cre/*loxP* strategy described in *Figure 1—figure supplement 3*.

We confirmed doxycycline-inducible expression of the full-length HaloTag fusion proteins using gel electrophoresis and fluorescent Halo ligands (*Figure 2—figure supplement 2*). The subcellular localization of both HaloTag fusions was then assessed using fluorescent Halo ligands. The PODXL fusion (henceforth referred to as apical-Halo) was restricted to the apical region of cells while the ATP1A1 fusion (henceforth referred to as basal-Halo) was restricted to the basal pole (*Figure 2A–B*). To assess the localization of alkynylated molecules produced by the Halo-seq protocol with these fusions, we visualized alkynylated molecules using in situ Click chemistry with a fluorescent azide as in *Figure 1—figure supplement 5*. We found that in cells expressing apical-Halo, alkynylated molecules were concentrated toward the apical poles of cells. Conversely, in cells expressing Basal-Halo, alkynylated molecules were concentrated toward the basal pole. In both cases, alkynylation was dependent upon the addition of DBF to cells (*Figure 2—figure supplements 3 and 4*). These results indicate that Halo-seq can spatially specifically alkynylate biomolecules across the apicobasal axis in epithelial monolayers, giving us confidence in the ability of Halo-seq to isolate RNA molecules that were localized to the apical and basal poles of epithelial cells.

We then confirmed the ability of the apical-Halo and basal-Halo fusions to facilitate RNA biotinylation using RNA dot blots (*Figure 2—figure supplement 5*) and purified the resulting products using streptavidin-coated beads. The apical-Halo and basal-Halo fusion proteins produced less biotinylated RNA than the cytoplasmic-Halo fusion. However, for both the apical and basal fusions, we recovered significantly more RNA following streptavidin pulldown from cells treated with DBF than from control untreated cells (*Figure 2—figure supplement 6*), further indicating the ability of these Halo fusions to mediate selective RNA biotinylation.

We isolated apically and basally localized RNA by performing Halo-seq in triplicate with apical-Halo and basal-Halo. We then quantified the resulting RNA samples using high-throughput sequencing of rRNA-depleted RNAseq libraries, calculating transcript and gene abundances using salmon and tximport (*Patro et al., 2017*; *Soneson et al., 2015*). Sequencing replicates of all Halo-seq samples first separate by genotype (PC1xPC2) then by fraction (PC2xPC3) in a PCA of gene expression profiles (*Figure 2—figure supplement 7*). To identify RNAs that were differentially localized across the apicobasal axis, we calculated an apical bias (AB) value for each gene. We defined this metric as the $\log_2$ ratio of a gene's abundance in the apical-Halo pulldown samples

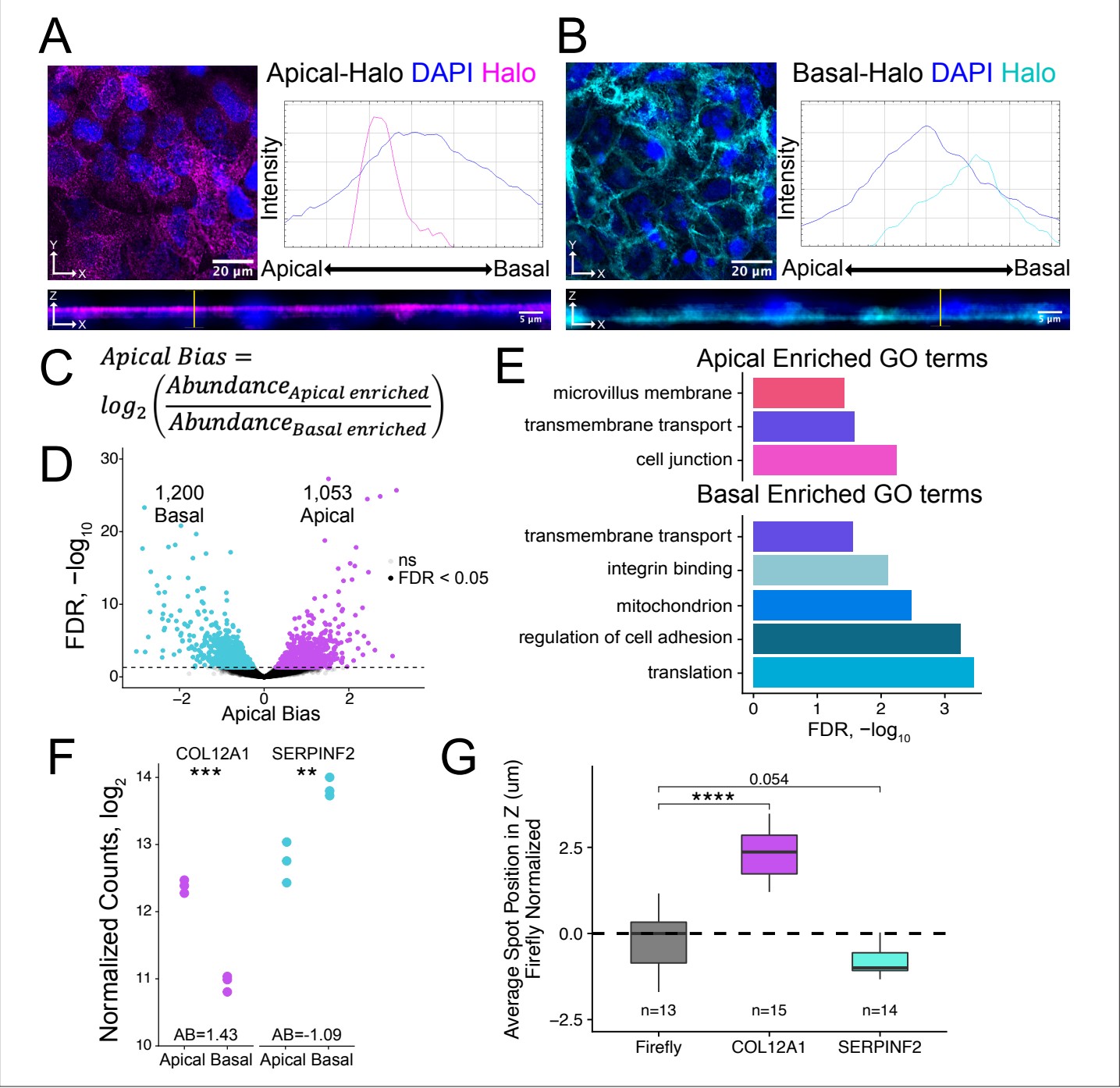

**Figure 2.** Halo-seq enriches RNAs at the apical and basal poles of enterocyte monolayers. (**A**) HaloTag protein fusions to PODXL and (**B**) ATP1A1 are localized to the apical and basal compartments respectively. HaloTag domains are visualized in fixed cells through the addition of a Halo ligand fluorophore shown in magenta (apical) and cyan (basal). DAPI stain marks the nuclei in blue. Profile lines across the XZ image demonstrate their apical or basal bias. Images are an average projection through the XZ axis of approximately 30 cells. (**C**) Equation for Apical Bias. RNA localization across the apicobasal axis is calculated with a metric termed Apical Bias as the $\log_2$ of a transcript's abundance in the apical enriched RNA fraction divided by its abundance in the basal enriched fraction. (**D**) Genes with differing Apical Bias following Halo-seq RNA labeling using the apical and basal-Halo fusions. FDR values calculated by DESeq2. (**E**) Enriched gene ontology (GO) terms associated with proteins encoded by RNAs identified as localized to the apical or basal pole of enterocytes. FDR calculated by topGO. (**F**) DESeq2-calculated normalized counts of candidate apical (COL12A1) and basal (SERPINF2) localized RNAs in the apical and basal enriched sequencing samples. p-Values calculated by DESeq2. (**G**) smFISH puncta position in Z. Firefly luciferase RNA included as a non-localized control. p-Values were calculated using a Wilcoxon rank-sum test. ns (not significant) represents p>0.05, * p<0.05, ** p<0.01, *** p<0.001 and **** represents p<0.0001.

*Figure 2 continued on next page*

*Figure 2 continued*

The online version of this article includes the following source data and figure supplement(s) for figure 2:

**Figure supplement 1.** C2bbe1 monolayers differentiated for 7 days on transwell inserts are highly polarized as demonstrated by immunofluorescence for Ezrin (apical) and NaK-ATPase (Basal) endogenous polarity protein markers.

**Figure supplement 2.** SDS-PAGE gel of doxycycline-induced apical-Halo and basal-Halo constructs (95 and 150 kDa respectively) visualized with Halo ligand fluorophore in red (JF 646).

**Figure supplement 2—source data 1.** GoPAGE Bis-Tris Precast Gel, 4–12% with Halo Ligand JF646 visualizing Halo-tagged proteins in doxycycline induced cell lysates.

**Figure supplement 2—source data 2.** GoPAGE Bis-Tris Precast Gel, 4–12% with coomassie visualizing total protein extracted from the halo tag expressing cell lysates.

**Figure supplement 3.** Cy3-Azide fluorophores attached to alkynylated biomolecules by Click chemistry visualize labeled molecules in fixed cells.

**Figure supplement 4.** Cy3-Azide fluorophores attached to alkynylated biomolecules by Click chemistry visualize labeled molecules in situ.

**Figure supplement 5.** in vitro biotinylation of RNA labeled in apical-Halo and basal-Halo expressing cells is dependent on DBF addition as visualized by streptavidin-HRP on an RNA dot blot.

**Figure supplement 5—source data 1.** RNA is spotted and crosslinked onto an Amersham Hybond-N +western blotting membrane.

**Figure supplement 5—source data 2.** RNA is spotted and crosslinked onto an Amersham Hybond-N +western blotting membrane.

**Figure supplement 6.** Percent enriched RNA after streptavidin pull-down of Apical-Halo and Basal-Halo-labeled RNAs.

**Figure supplement 7.** Principal component analysis of gene expression values from the cytoplasmic-Halo, apical-Halo, and basal-Halo experiments.

**Figure supplement 8.** Mitochondria are visualized with both MitoView Green and immunofluorescence for TOM20 protein in fixed cells.

**Figure supplement 9.** Direct comparison of Halo-seq calculated Apical Bias in Human C2bbe1 monolayers and Laser Capture Microdissection (LCM) calculated Apical Bias in adult Mouse enterocytes (*Moor et al., 2017*).

**Figure supplement 10.** Representative smFISH images in fixed cells.

**Figure supplement 11.** Number of smFISH puncta per cell.

**Figure supplement 12.** Transcripts per million for each candidate localized RNA in each sequenced input sample (n=9).

**Figure supplement 13.** Cumulative distribution of smFISH spots across their position in Z as calculated by FISH-quant.

compared to its abundance in the basal-Halo pulldown samples (*Figure 2C*, *Supplementary file 1*). Genes with positive AB values are therefore apically-enriched while those with negative AB values are basally-enriched. AB values and their associated p values were calculated using DESeq2 (*Love et al., 2014*). With this approach, transcripts from 1053 genes were found to be significantly apically localized (FDR < 0.05) while transcripts from 1200 genes were significantly basally localized (*Figure 2D*). These results suggest that many RNAs are asymmetrically distributed across the apico-basal axis in epithelial cells.

To preliminarily characterize the apical and basal transcriptomes, we used TopGO (*Alexa, 2010*) to identify enriched gene ontology (GO) terms associated with the proteins encoded by RNAs localized to each compartment (*Figure 2E*). The term 'trans-membrane transport' was significantly associated with RNAs localized to both compartments (p<0.05), consistent with the fact that both the apical and basal membranes of enterocytes play major roles in nutrient import and export (*Kiela and Ghishan, 2016*; *Snoeck et al., 2005*). Unique GO terms associated with apically localized RNAs included 'cell junctions' including apical tight junctions, and 'microvillus membrane' which is exclusively associated with enterocyte apical membrane (*Crawley et al., 2014*; *Sluysmans et al., 2017*). Proteins encoded by basally localized RNAs were enriched (p<0.05) for functions associated with cell adhesion and integrin binding, both of which are important for basal membrane anchoring to the extracellular matrix (*Yurchenco, 2011*).

RNAs from nuclear-encoded mitochondrial proteins were also found to be basally enriched. Because these RNAs are often localized to mitochondria due to their translation on the mitochondrial surface (*Corral-Debrinski et al., 2000*; *Williams et al., 2014*), this would imply that mitochondria themselves may also be basally localized in C2bbe1 monolayers. To test this, we visualized mitochondria using both immunofluorescence and a mitochondrial-specific dye, mitoView. Both methods indicated that the localization of mitochondria was, in fact, basally biased (*Figure 2—figure supplement 8*). These results gave us confidence in the reliability of the Halo-seq quantifications.

## Halo-seq results are consistent with previous RNA localization datasets in polarized enterocytes

A previous study used laser capture microdissection (LCM) and RNAseq to assess RNA localization across the apicobasal axis in adult mouse intestine tissue slices (*Moor et al., 2017*). In this study, RNA samples from apical and basal LCM slices were compared to define RNAs differentially distributed across the apicobasal axis of these cells. To assess the quality of our Halo-seq data, we compared our AB values to those defined in this dataset. Using the mouse intestine data, we calculated AB values for all genes and compared them to Halo-seq derived AB values of their human orthologs. We found a modest (*R*=0.08), yet positive and significant (p=0.003) correlation between these two datasets (*Figure 2—figure supplement 9*). This correlation overcomes differences in samples (cultured cells vs. tissue), methodology (Halo-seq vs. LCM), and species (human vs. mouse). We found this correlation to be encouraging and further suggestive of the ability of Halo-seq to faithfully report on RNA localization across the apicobasal axis in epithelial cells.

## Validation of Halo-seq results using single-molecule RNA FISH

To experimentally validate Halo-seq's ability to identify transcripts differentially localized across the apicobasal axis, we performed single molecule fluorescent in situ hybridization (smFISH) to visualize endogenous transcripts within C2bbe1 monolayers (*Tsanov et al., 2016*). We selected one transcript with a positive AB value, *COL12A1*, and one transcript with a negative AB value, *SERPINF2,* for validation (*Figure 2F*, *Figure 2—figure supplement 10*). Spots corresponding to RNA molecules in smFISH images were detected using FISH-quant (*Mueller et al., 2013*; *Tsanov et al., 2016*), and the 3D coordinates of each spot were identified. This allowed calculation of the position of each transcript's position along the image's Z axis, which corresponds to the cell's apicobasal axis. When analyzing smFISH images for each endogenous transcript, we ensured that the number of spots detected per cell was proportional to the expression levels of the transcript as calculated by RNAseq (*Figure 2—figure supplements 11 and 12*), suggesting that each custom designed smFISH probe set (*Tsanov et al., 2016*) was faithfully detecting its cognate transcript. By comparing to the localization of an exogenous transcript encoding Firefly luciferase as a nonlocalized control, we found that *COL12A1* transcripts were significantly apically localized while *SERPINF2* transcripts were basally localized (*Figure 2G*, *Figure 2—figure supplement 13*). These results are consistent with their AB values as determined by Halo-seq, giving us additional confidence in the reliability of the Halo-seq dataset.

## Pyrimidine-rich motifs in 5′ UTRs are sufficient to drive RNA transcripts to the basal pole of epithelial monolayers

Using our Halo-seq derived dataset of RNAs localized across the apicobasal axis of enterocytes, we observed that mRNAs encoding ribosomal proteins (RP mRNAs) were, as a class, strongly basally localized (*Figure 3A*). This is consistent with previous observations made in adult mouse intestines (*Moor et al., 2017*), as well as with a LCM-based study of RNA localization in *Drosophila* follicular epithelial cells (*Cassella and Ephrussi, 2022*; *Figure 3—figure supplement 1*), further suggesting that our model and technique are faithfully recapitulating phenomena that occur in vivo. While RP mRNAs are known to be basally localized in epithelial cells (*Cassella and Ephrussi, 2022*; *Moor et al., 2017*), the mechanisms underlying this localization are unknown.

Essentially all RP mRNAs share a highly conserved, pyrimidine-rich sequence motif in their 5′ UTR sometimes called a 5′ TOP (terminal oligopyrimidine) motif that likely regulates their translation (*Iadevaia et al., 2008*; *Lahr et al., 2017*; *Yoshihama et al., 2002*). Given that RP mRNAs were strongly basally localized in epithelial cells, we hypothesized that 5′ TOP motifs may be regulating their localization. Similar 5′ UTR motifs called pyrimidine-rich translational elements (PRTEs) have additionally been found to regulate the translation of some non-RP mRNAs (*Hsieh et al., 2012*). We found that PRTE-containing RNAs were also basally enriched, suggesting that like 5′ TOP motifs, PRTEs may also be regulating RNA localization (*Figure 3A*). Although there is some inconsistency in the literature, in general, 5′ TOP motifs are defined as being located *immediately* next to the 5′ cap (*Berman et al., 2021*) while PRTEs can be more generally localized anywhere within the 5′ UTR. There are conflicting reports as to whether the proximity of a pyrimidine-rich motif to the 5′ cap affects its function (*Al-Ashtal et al., 2021*; *Hong et al., 2017*; *Lahr et al., 2017*).

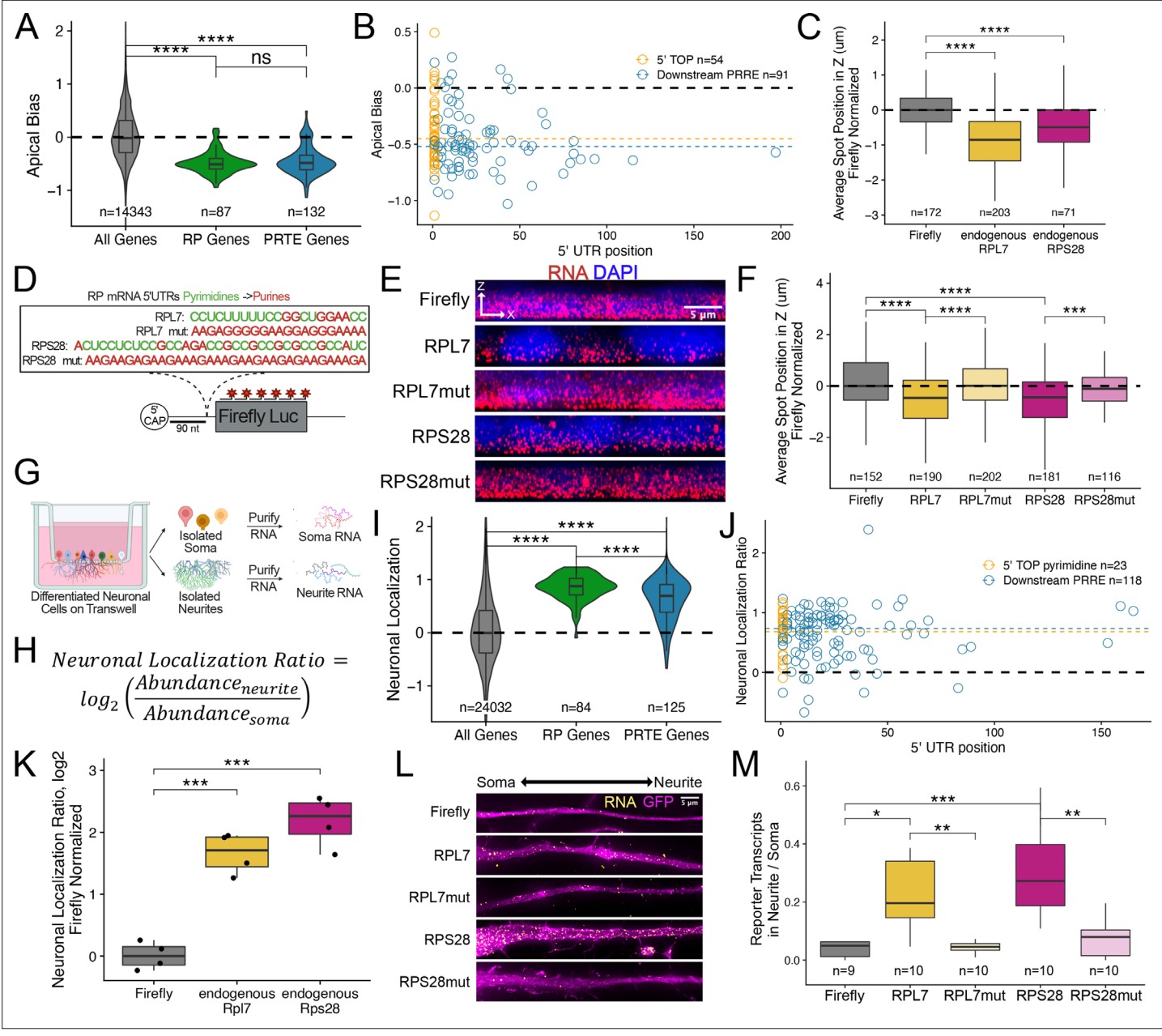

**Figure 3.** Ribosomal protein mRNAs are localized in a PRRE-dependent manner in multiple cell types. (**A**) Apical Bias for ribosomal protein (RP) genes and PRTE-containing genes as compared to all genes. P-values were calculated using a Wilcoxon rank-sum test. (**B**) Apical Bias for transcripts containing pyrimidine-rich 5′ UTR elements as a function of the position of the element within the 5′ UTR. Dotted lines represent medians for each class. (**C**) Localization of endogenous RPL7 and RPS28 transcripts in C2bbe1 monolayers as determined by smFISH. Values represent the average position per cell. Numbers of cells interrogated for each transcript are indicated. (**D**) Schematic of ribosomal protein PRRE reporter constructs. Mutant versions were created by swapping each pyrimidine for a purine. Motifs were inserted roughly 90 nucleotides downstream of the transcription start site. (**E**) Representative images of smFISH spots for each PRRE reporter construct. RNA signal shown in red while DAPI stains mark the nuclei blue. Images are a max projection through the XZ axis of roughly 20 cells. (**F**) smFISH puncta position in Z normalized to median untagged Firefly luciferase transcript position. P-values were calculated using a Wilcoxon rank-sum test. Values represent the average position per cell. Numbers of cells interrogated for each construct are indicated. (**G**) Schematic of neuronal subcellular fractionation and localized RNA isolation. (**H**) Equation for Neuronal Localization Ratio. Neuronal RNA localization is calculated with a metric termed Neuronal Localization Ratio as the $\log_2$ of a transcript's abundance in the neurite fraction divided by its abundance in the soma fraction. (**I**) Neuronal Localization Ratio (LR) for ribosomal protein genes and PRTE-containing genes as compared to all genes. p-Values were calculated using a Wilcoxon rank-sum test. (**J**) Neurite localization ratio for transcripts containing pyrimidine-rich 5′ UTR elements as a function of the position of the element within the 5′ UTR. Dotted lines represent medians for each class. (**K**) Neurite-enrichment of endogenous Rpl7 and Rps28 transcripts as quantified by RT-qPCR following neuronal fractionation. Endogenous gene expression normalized to

*Figure 3 continued on next page*

*Figure 3 continued*

the endogenous RNA Tsc1. (**L**) Representative images of smFISH for each PRRE reporter construct. Images are a max projection through the XY axis of a single neurite positioned with the soma to the left. RNA signal shown in yellow while cell outlines marked by GFP signal shown as magenta. (**M**) Number of smFISH puncta in neurites normalized to soma. For all panels, p-values were calculated using a Wilcoxon rank-sum test. Numbers of cells interrogated for each construct are indicated. ns (not significant) represents p>0.05, * p<0.05, ** p<0.01, *** p<0.001 and **** represents p<0.0001.

The online version of this article includes the following figure supplement(s) for figure 3:

**Figure supplement 1.** Localization of ribosomal protein mRNAs across the apicobasal axis of adult mouse enterocytes (*Moor et al., 2017*), the apicobasal axis of *Drosophila* follicular cells (*Cassella and Ephrussi, 2022*) and in migrating cell projections (*Dermit et al., 2020*) according to various sequencing experiments.

**Figure supplement 2.** Genes with differing Apical Bias with candidate RP mRNAs highlighted.

**Figure supplement 3.** Representative smFISH images for endogenous RPL7 and RPS28 transcripts in an orthogonal, XZ projection of roughly 20 cells.

**Figure supplement 4.** Transcript abundance per cell for endogenous RPL7 and RPS28 transcripts as measured by smFISH.

**Figure supplement 5.** Transcript abundance per cell for PRRE-containing reporters in C2bbe1 cells as measured by smFISH.

**Figure supplement 6.** PRRE reporter transcript localization in HCA-7 monolayers.

**Figure supplement 7.** Transcript abundance per cell for PRRE-containing reporters in HCA7 cells as measured by smFISH.

**Figure supplement 8.** PRRE reporter construct localization in MDCK monolayers.

**Figure supplement 9.** Transcript abundance per cell for PRRE-containing reporters in MDCK cells as measured by smFISH.

**Figure supplement 10.** Neuronal localization ratio of all genes with candidate RP mRNAs highlighted.

**Figure supplement 11.** Human and mouse 5′ UTR alignments for RPL7 and RPS28.

**Figure supplement 12.** Total smFISH reporter construct puncta detected in neurites and soma by FISH-quant.

**Figure supplement 13.** Enrichment of PRRE-containing reporter transcripts in neurite RNA fractions as determined by RT-qPCR.

To determine if the location of pyrimidine-rich elements within 5′ UTRs of mRNAs was connected to RNA localization, we defined the presence and location of pyrimidine-rich elements within all RP and PRTE-containing mRNAs (see Methods). If a UTR contained multiple pyrimidine-rich elements, we used the position of the most 5′ element. We observed no difference in basal localization between RNAs that contained 5′ TOP motifs immediately adjacent to the 5′ cap and those that contained PRTE elements further downstream in the 5′ UTR (*Figure 3B*). From this, we conclude that there is no relationship between the localization of an RNA and the *position* of pyrimidine-rich elements within its 5′ UTR. Therefore, we will refer to these elements as pyrimidine-rich regulatory elements (PRREs) as (1) their position within 5′ UTRs seems unimportant and (2) they may regulate more than translation.

Although the above analyses showed a relationship between PRREs and RNA localization, they do not demonstrate causality. To directly test the ability of PRREs to regulate RNA localization, we selected two RP mRNAs, *RPL7* and *RPS28*, that were basally localized in the Halo-seq data (*Figure 3— figure supplement 2*) and validated their localization using smFISH (*Figure 3C*, *Figure 3—figure supplements 3 and 4*, *Supplementary file 2*). We then placed their entire 5′ UTRs in front of the open reading frame (ORF) of Firefly luciferase to create two reporter transcripts such that the PRREs in these transcripts were approximately 90 nt from the 5′ cap (*Figure 3D*). As negative controls, we created similar reporter RNAs containing mutated PRREs such that every pyrimidine within the motif was replaced with a purine (*Figure 3D*). We determined the localization of these reporter RNAs in C2bbe1 intestinal epithelial monolayers using smFISH probes designed to target the Firefly luciferase ORF.

On its own, the Firefly luciferase transcript is an exogenous, non-localized RNA. We found that addition of the PRREs from both *RPL7* and *RPS28* were sufficient to drive the Firefly luciferase transcript to the basal compartment. Mutated PRREs were not able to drive basal localization (*Figure 3E*). We then quantified the localization of each PRRE reporter's transcripts and found that PRRE-containing reporter transcripts were significantly localized to the basal pole of the cells (p<0.01, Wilcoxon rank sum test) while those containing mutant PRREs were not (*Figure 3F*). To our knowledge, this is the first identified RNA element that is sufficient to drive RNAs to the basal pole of epithelial cells.

5′ TOP motifs have been previously shown to negatively regulate RNA stability (*Fonseca et al., 2015*). Consistent with this, mutant PRRE reporter transcripts were more abundant than their wild-type counterparts (*Figure 3—figure supplement 5*). It should be noted again, though, that in these reporters the PRREs are located well into the 5′ UTR, further demonstrating that they need not be immediately adjacent to the 5′ cap to exert their potential regulatory ability.

To more broadly assess the ability of the PRRE motif to regulate RNA localization in intestinal epithelial cells, we investigated the localization of our reporter constructs in a different human intestinal enterocyte cell line, HCA-7 cells. HCA-7 cells create monolayers that are more colon-like than the enterocyte-like C2bbe1 monolayers (*Kirkland, 1985*). Reporter constructs were integrated into HCA-7 lox cells via cre-*lox* recombination. We again found that PRRE motifs from RPL7 and RPS28 were both sufficient for basal RNA localization in HCA-7 monolayers while mutated PRRE motifs were not (*Figure 3—figure supplement 6*). Also, as in the C2bbe1 monolayers, PRRE motifs were associated with decreased transcript abundance compared to their mutated counterparts (*Figure 3—figure supplement 7*).

To test the ability of PRRE motifs to regulate RNA localization in epithelial cells generally, we integrated our human PRRE reporter constructs into MDCK cells through *cre*-mediated recombination. MDCK cells are epithelial cells derived from canine kidneys and have been used to study polarity across the apicobasal axis (*Balcarova-Ständer et al., 1984*). As before, we found that PRRE-containing reporter RNAs were strongly basally localized while mutant PRRE motifs were not able to drive basal RNA localization (*Figure 3—figure supplement 8*) and that PRREs were associated with decreased transcript abundance (*Figure 3—figure supplement 9*). PRRE motifs are therefore able to promote basal RNA localization in a variety of epithelial cell types across multiple mammalian species. Furthermore, elements derived from human transcripts are able to regulate RNA localization in canine-derived cells.

## PRRE motifs regulate RNA localization in morphologically dissimilar cell types

RP mRNAs have been found to be localized to cellular projections made by neurons and migrating cancer cells (*Dermit et al., 2020*; *Shigeoka et al., 2019*; *Taliaferro et al., 2016*; *Figure 3—figure supplement 1*). These observations were made by sequencing RNA fractions collected following mechanical fractionation using microporous membranes where small projections grow through pores restricting cell bodies on top of the membrane (*Figure 3G*). In these experiments, RNA localization can be described as a localization ratio (LR) comparing the $\log_2$ abundance of a transcript in projection or neurite fraction to its abundance in the soma or cell body fraction (*Figure 3H*). To identify RNAs robustly localized into neurites, we combined published datasets from over 30 unique experiments where neuronal cells were subjected to mechanical fractionation. This dataset included neuronal samples spanning a range of neuronal cell types from cultured cell lines to primary cortical neurons (*Goering et al., 2020*; *Taliaferro et al., 2016*). For each neuronal sample, we calculated an LR value for each gene in each sample. These were then Z-normalized within a sample, and the gene's LRz value was defined as the gene's median normalized LR value across all samples. Genes with positive LRz values therefore have RNAs that are neurite-enriched while genes with negative LRz values have RNAs that are cell body enriched. Using this dataset, we confirmed that RP mRNAs are efficiently localized to neurites across dozens of experiments (*Figure 3I*).

As in the epithelial cells, RNAs previously defined as having a pyrimidine-rich translation element (PRTE) (*Hsieh et al., 2012*) were also neurite-enriched, and the location of pyrimidine-rich elements with 5′ UTRs was not correlated with RNA localization (*Figure 3J*). This again indicates that PRREs can likely regulate RNA localization in neurons independent of their proximity to the 5′ cap.

Although RP mRNAs are enriched in neuronal projections, the RNA sequences within the RP mRNAs that regulate this localization are unknown. We reasoned that because PRRE motifs are highly conserved and had the ability to regulate RNA localization in epithelial cells that perhaps they may similarly be able to drive RNA transcripts to neurites. Endogenous mouse Rpl7 and Rps28 transcripts were found to be localized to neurites in the transcriptome-wide subcellular fractionation datasets (*Figure 3—figure supplement 10*) and by RT-qPCR of CAD cell body and neurite RNA samples (*Figure 3K*). The PRRE motifs of these mouse RNAs differ from their human orthologs by only a few nucleotides (*Figure 3—figure supplement 11*).

To test if PRRE elements are sufficient for neurite localization, we integrated and expressed our human PRRE reporter RNAs derived from human RPL7 and RPS28 in CAD mouse neuronal cells. Using smFISH, we found that reporters with wildtype PRRE motifs were efficiently localized to neurites while reporters that contained mutated PRRE motifs or lacked PRRE motifs were not (*Figure 3L*). We quantified the smFISH data by comparing the number of RNA puncta in the neurite to the number

of puncta in the soma for each reporter. We found that PRRE-containing reporters were significantly more enriched in neurites than reporters that contained mutated PRRE motifs (p<0.001, Wilcoxon rank-sum test) (*Figure 3M*). This effect was driven by a neurite-specific increase in transcript abundance and not due to any cell-wide RNA expression changes (*Figure 3—figure supplement 12*).

We then confirmed this result using an orthologous technique. We mechanically fractionated reporter-expressing CAD cells into cell body and neurite fractions and quantified reporter RNA in both fractions using qRT-PCR. Consistent with the smFISH results, we found that PRRE-containing reporter RNAs were neurite-enriched while RNAs containing mutated PRREs were significantly less enriched (*Figure 3—figure supplement 13*).

Human PRRE motifs are therefore sufficient for neurite localization in mouse neuronal cells, suggesting that PRRE-mediated RNA localization mechanisms are conserved across species. Further, given that PRRE motifs regulate RNA localization in intestinal epithelial cells, neuronal cells, and migrating cancer cells (*Dermit et al., 2020*; *Figure 3F and M*, *Figure 3—figure supplement 1*), these results demonstrate that PRRE-mediated RNA localization mechanisms are conserved across cell types with vastly different morphologies.

## Neurite localization driven by PRRE motifs is Larp1 dependent

We next sought to identify RBPs that are required for PRRE-mediated RNA transport in neuronal cells. Multiple members of the highly conserved La superfamily of RBPs (also known as La Related Proteins or LARPs) regulate multiple aspects of RNA metabolism in a variety of cell types (*Deragon, 2020*; *Dock-Bregeon et al., 2021*). Two of these, Ssb and Larp1, have been previously found to bind pyrimidine-rich motifs within 5′ UTRs (*Dermit et al., 2020*; *Lahr et al., 2015*; *Maraia et al., 2017*). Given this, we assayed the requirement of Larp family members for PRRE-mediated RNA localization in CAD cells by performing a targeted RNAi screen and monitoring the localization of our PRRE-containing reporters. Importantly, although Larp6 is required for endogenous RP mRNA transport in migrating cancer cell projections (*Dermit et al., 2020*), it is not expressed in CAD cells and therefore was not included in the screen.

We knocked down Larp family members Ssb, Larp1, Larp4, and Larp7 using siRNA transfection. Knockdown efficiencies assayed by qPCR (*Figure 4—figure supplement 1*) and immunoblotting (*Figure 4A*) were between 80% and 95% for each Larp protein assayed. We hypothesized that if a Larp RBP was responsible for PRRE-mediated RNA localization to neurites, we would see a decrease in the neurite localization of PRRE-containing reporters when expression of the Larp RBP was reduced. Following the depletion of Larp1, we observed that wildtype PRRE-containing reporters were significantly less neurite-enriched, demonstrating that Larp1 is required for their efficient transport. This effect was PRRE-specific as the localization of mutant PRRE-containing reporters was unaffected (*Figure 4B–C*). Larp1 was the only assayed RBP that was required for PRRE-mediated localization as knockdown of the other Larp family members did not consistently affect the localization of wild-type PRRE reporters (*Figure 4—figure supplement 2*). Larp1 therefore mediates PRRE localization to neurites, and this function is not shared with other Larp family members.

## Larp1 depletion reduces number and length of neurites

During our Larp depletion experiments, we observed that the loss of Larp1 resulted in CAD cells with defects in neurite outgrowth (*Figure 4—figure supplement 3*). Specifically, the loss of Larp1 expression reduced both the number of cells producing neurites (*Figure 4—figure supplement 4*) and shortened overall neurite length (*Figure 4—figure supplement 5*). Larp1 was the only tested Larp protein for which this effect was observed. The depletion of other Larp proteins increased the number of cells producing neurites (*Figure 4—figure supplement 4*), and in the case of Larp4, also increased overall neurite length (*Figure 4—figure supplement 5*). Larp1 is therefore both required for transport of PRRE-containing transcripts to neurites and required for proper neurite outgrowth. However, because many Larp proteins, including Larp1, are implicated in a wide range of RNA regulatory modes, we cannot directly ascribe this neurite phenotype to the loss of RP mRNA localization.

## Basal localization of PRRE motifs in epithelial cells is LARP1 dependent

Following our knockdown experiments in mouse neuronal cells, we suspected that the pyrimidine-binding protein LARP1 may be responsible for the basal localization of PRRE-containing RNAs in

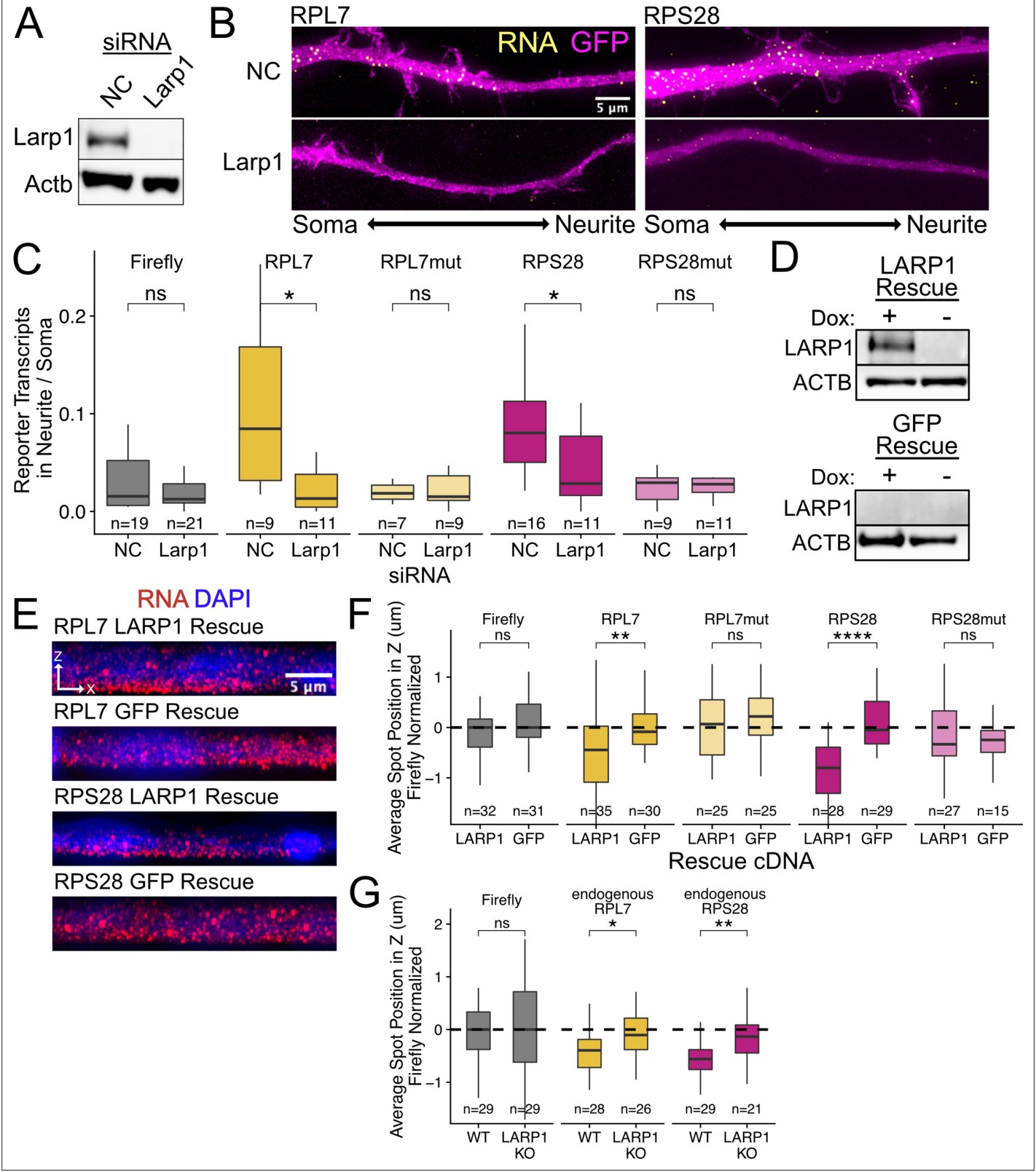

**Figure 4.** PRRE localization is mediated by LARP1 in enterocyte monolayers and neurons. (**A**) Immunoblot for Larp1 and Actb showing knockdown of Larp1 protein in neuronal cells when treated with Larp1 siRNAs as compared to a negative control (NC) siRNA. (**B**) Representative images of smFISH for each wildtype PRRE reporter construct treated with Larp1 siRNAs or NC siRNAs. Images are a max projection through the XY axis of a single neurite positioned with the soma to the left. RNA signal shown in yellow while cell outlines marked by GFP signal shown as magenta. (**C**) Number of smFISH

*Figure 4 continued on next page*

*Figure 4 continued*

reporter transcript puncta in neurites normalized to soma. p-Values were calculated using a Wilcoxon rank-sum test. Numbers of cells interrogated for each construct are indicated. (**D**) Immunoblot for LARP1 and ACTB showing knockout and doxycycline-induced rescue of LARP1 levels in C2bbe1 LARP1 knockout cells. (**E**) Representative images of smFISH for each wildtype PRRE reporter construct. RNA signal shown in red while DAPI marks the nuclei in blue. Images are a max projection through the XZ axis of many cells. (**F**) smFISH puncta position in Z of the indicated Firefly luciferase reporters normalized to median untagged Firefly luciferase transcript position. p-Values were calculated using a Wilcoxon rank-sum test. Values represent the average value per cell. Numbers of cells interrogated for each construct are indicated. (**G**) smFISH puncta position in Z of the indicated endogenous transcripts normalized to median untagged Firefly luciferase transcript position. Vp-alues were calculated using a Wilcoxon rank-sum test. Values represent the average value per cell. Numbers of cells interrogated for each transcript are indicated. ns (not significant) represents p>0.05, * p<0.05, ** p<0.01, *** p<0.001 and **** represents p<0.0001.

The online version of this article includes the following source data and figure supplement(s) for figure 4:

**Source data 1.** Protein lysates from siRNA treated CAD cells were ran on a GoPAGE Bis-Tris Precast Gel, 4–12% before being transferred to a PVDF membrane via the iBlot dry transfer device.

**Source data 2.** The ladder from the same blot was imaged.

**Source data 3.** Protein lysates from siRNA-treated CAD cells were ran on a GoPAGE Bis-Tris Precast Gel, 4–12% before being transferred to a PVDF membrane via the iBlot dry transfer device.

**Source data 4.** The ladder from the same blot was imaged.

**Source data 5.** Protein lysates from C2bbe1 LARP1 Knockout cells rescued with doxycycline-inducible cDNAs were ran on a GoPAGE Bis-Tris Precast Gel, 4–12% before being transferred to a PVDF membrane via the iBlot dry transfer device.

**Source data 6.** Protein lysates from C2bbe1 LARP1 Knockout cells rescued with doxycycline-inducible cDNAs were ran on a GoPAGE Bis-Tris Precast Gel, 4–12% before being transferred to a PVDF membrane via the iBlot dry transfer device.

**Source data 7.** The ladder from the same blot was imaged.

**Figure supplement 1.** Percent knockdown of Larp RNAs by targeting and negative control siRNAs as measured by qPCR normalized to HPRT.

**Figure supplement 2.** Reporter localization in neurons treated with different Larp siRNAs.

**Figure supplement 3.** Representative brightfield images of Larp1 and negative control siRNA-treated differentiated CAD cells.

**Figure supplement 4.** Proportion of cells with or without defined neurites when treated with different Larp siRNAs.

**Figure supplement 5.** Neurite length of multiple Larp siRNA-treated cells normalized to negative control (NC) siRNA-treated cells.

**Figure supplement 6.** LARP1 CRISPR/Cas9 knockout Strategy.

**Figure supplement 7.** PCR amplification of the locus using primers that flank both gRNA cut sites shows loss of the sequence between exons 2 and 19.

**Figure supplement 7—source data 1.** 1% Agarose gel of PCR products using primers flanking Exon2 of LARP1 as well as the complete deletion of LARP1 by CRISPR/Cas9 targeting gRNAs in Exon2 and 19.

**Figure supplement 8.** Relative expression of LARP1 rescue lines as normalized to HPRT then normalized to wildtype C2bbe1 LARP1 expression.

**Figure supplement 9.** Expression of PRRE-containing reporter transcripts in LARP1 knockout and rescue cells as assayed by smFISH.

**Figure supplement 10.** Localization of endogenous RPL7 and RPS28 transcripts in wildtype and LARP1 knockout C2bbe1 monolayers as assayed by smFISH.

**Figure supplement 11.** Transcript abundance of endogenous RPL7 and RPL28 transcripts in wildtype and LARP1 knockout C2bbe1 monolayers as assayed by smFISH.

human epithelial monolayers. To test this, we used CRISPR/Cas9 to generate a LARP1 knockout C2bbe1 cell line.

Using two guide RNAs, we targeted exon 2 and exon 19, both of which are included in all major isoforms of LARP1, with the goal of deleting the sequence in between them (*Figure 4—figure supplement 6*). Because C2bbe1 cells may not be diploid at the *LARP1* locus, we expected to identify multiple Cas9-generated *LARP1* alleles in a single clonal line. We identified one clonal line in which no wild type *LARP1* was detectable. One or more alleles had lost the sequence in between exon 2 and exon 19 and the only remaining allele contained a 53 base pair deletion in exon 2 that resulted in a premature stop codon (*Figure 4—figure supplements 6 and 7*). The *LARP1* RNA produced by this clone is therefore not expected to produce a full length protein product and should be targeted for nonsense-mediated decay (NMD) (*Figure 4—figure supplement 6*). Consistent with this, the level of *LARP1* RNA produced in this clone was less than 3% of that produced in wildtype cells (*Figure 4—figure supplement 8*).

To reintroduce expression of LARP1 in the selected LARP1 knockout clone, we integrated a full-length LARP1 rescue cDNA under a doxycycline-inducible promoter via *cre*-mediated recombination.

This LARP1 KO-rescue line expressed LARP1 protein only in the presence of doxycycline. As a control, we also created a line in which the knockout was rescued with doxycycline-inducible GFP (*Figure 4D*).

By expressing our PRRE-containing reporter RNAs in the LARP1 knockout and rescue lines, we found that PRRE-containing reporters were basally localized only when LARP1 was present, indicating that LARP1 is required for their localization (*Figure 4E and F*). We wondered if the loss of LARP1 expression would also reduce the basal localization of endogenous RPL7 and RPS28 transcripts. Using smFISH probes targeting the endogenous transcripts (*Supplementary file 2*), we assayed the localization of the endogenous RP RNAs across the apicobasal axis in our wildtype C2bbe1 cells as well as in our LARP1 KO cell line (*Figure 4—figure supplements 9 and 10*). We found that loss of LARP1 did not change the abundance of endogenous RP transcripts (*Figure 4—figure supplement 11*). However, the basal localization of the endogenous transcripts was reduced when LARP1 was not present (*Figure 4G*). Altogether, we found that LARP1 mediates the basal localization of PRRE-containing RNAs in epithelial monolayers and the neurite localization of the same RNAs in neuronal cells.

## PRRE motifs must be in the 5′ UTR of transcripts in order to mediate RNA localization

To better understand how PRRE motifs were regulating the localization of our reporter transcripts, we asked if the location of the motif in the transcript beyond the 5′ UTR changed its ability to regulate RNA localization. Much of the data demonstrating that LARP1 interacts with pyrimidine-rich elements does so with motifs that are close to the 5′ 7-methylguanosine cap (*Al-Ashtal et al., 2021*; *Lahr et al., 2017*; *Philippe et al., 2018*). To test if PRRE motifs could regulate RNA localization when they were in 3′ UTRs, we placed the RPL7 and RPS28 PRRE motifs after the Firefly luciferase ORF, thus creating 3′ PRRE reporter constructs (*Figure 5A*). We reasoned that if the PRREs must be in 5′ UTRs in order to regulate RNA localization, the 3′ PRRE reporters would display a reduced ability to regulate RNA localization. In C2bbe1 monolayers, we found that 3′ PRRE reporters were significantly less basally localized than their 5′ PRRE counterparts (*Figure 5B*). We observed similar results in mouse neuronal cells, finding that 3′ PRRE reporters were significantly less efficient at driving RNAs to neurites than their 5′ PRRE counterparts (*Figure 5C*). Therefore, in order to efficiently regulate RNA localization, PRRE motifs must be in the 5′ UTR of transcripts. This is consistent with LARP1 binding both PRRE motifs and the 5′ 7-methylguanosine cap (*Al-Ashtal et al., 2021*; *Lahr et al., 2017*; *Philippe et al., 2018*).

Interestingly, the mutant, purine-rich 3′ PRRE reporter transcripts were strongly basally localized in epithelial cells (*Figure 5—figure supplements 1 and 2*). We found similar results in neuronal cells where the mutant RPL7 3′ PRRE reporter RNA was neurite-localized (*Figure 5—figure supplements 3 and 4*). Computational prediction of RNA secondary structure revealed that the mutant RPL7 PRRE motif is likely folded into a G-quadruplex conformation (*Lorenz et al., 2011 Figure 5—figure supplement 5*). G-quadruplex structures within 3′ UTRs have been previously found to target RNAs to neurites (*Goering et al., 2020*; *Subramanian et al., 2011*) and may therefore be similarly regulating RNA localization in epithelial cells.

To ask if LARP1 regulates the localization of 3′ PRRE-containing transcripts, we assayed 3′ PRRE localization in LARP1- and GFP-rescued LARP1 KO C2bbe1 monolayers using smFISH. We found that the loss of LARP1 did not affect the localization of 3′ PRRE-containing transcripts (*Figure 5D*, *Figure 5—figure supplements 6 and 7*). Similarly, in mouse neuronal cells, siRNA-mediated depletion of Larp1 did not affect the neurite localization of 3′ PRRE-containing transcripts (*Figure 5E*, *Figure 5—figure supplements 8 and 9*). These results further strengthen the argument that LARP-1-mediated RNA localization requires a PRRE motif in the 5′ UTR of the transcript.

## An RNA element within the 3′ UTR of Net1 is necessary and sufficient for RNA localization in neuronal and epithelial cells

We had identified an RNA element, PRRE motifs, that regulated RNA localization in both epithelial and neuronal cells. We wondered whether other RNA elements may be similarly able to regulate RNA localization in both cell types. We therefore set out to find other examples of RNAs that were localized in both cell types as well as the sequences that drive their localization.

*Net1* RNA encodes a guanine exchange factor and is strongly localized to neurites (*Arora et al., 2022a*), the basal pole of intestinal epithelial cells (*Moor et al., 2017*), and to the projections of

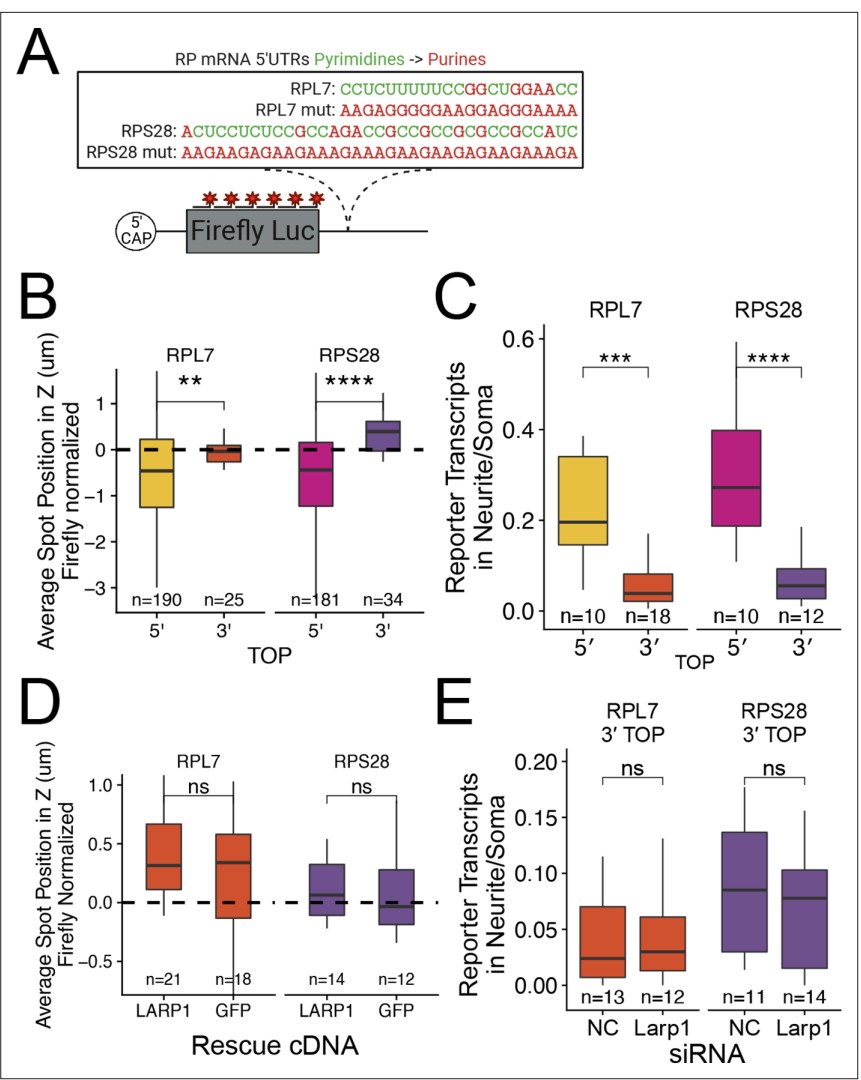

**Figure 5.** PRRE motifs must be in the 5' UTR for efficient transcript localization. (**A**) Schematic of ribosomal protein 3' PRRE reporter constructs. Mutant versions were created by swapping each pyrimidine for a purine. PRRE motif sequences are identical to 5' PRRE reporter constructs. (**B**) 3' PRRE reporter transcript localization in C2bbe1 monolayers. smFISH puncta position of the indicated Firefly luciferase reporters in Z normalized to median untagged Firefly transcript position. p-Values were calculated using a Wilcoxon rank-sum test. Values represent the average value per cell. (**C**) 3' PRRE construct localization in neurons. Number of reporter RNA smFISH puncta in neurites normalized to soma. p-Values were calculated using a Wilcoxon rank-sum test. (**D**) 3' PRRE construct localization in LARP1 knockout and rescued C2bbe1 monolayers. smFISH puncta position of the indicated Firefly luciferase reporters in Z normalized to median untagged Firefly luciferase transcript position. p-Values calculated using a Wilcoxon rank-sum test. Values represent the average value per cell. (**E**) 3' PRRE construct localization in neurons depleted of Larp1 by siRNA. Number of reporter RNA smFISH puncta in neurites normalized to soma. p-Values calculated using a Wilcoxon rank-sum test. ns (not significant) represents p>0.05, * p<0.05, ** p<0.01, *** p<0.001 and **** represents p<0.0001. For B-E, Numbers of cells interrogated for each construct are indicated.

The online version of this article includes the following figure supplement(s) for figure 5:

**Figure supplement 1.** Transcript abundance of 3' PRRE reporter transcripts in C2bbe1 monolayers as calculated by smFISH.

**Figure supplement 2.** 3' PRRE reporter transcript localization in C2bbe1 monolayers.

**Figure supplement 3.** Transcript abundance of 3' PRRE reporter transcripts in the neurite and soma compartments of CAD cells as calculated by smFISH.

**Figure supplement 4.** Number of 3' PRRE reporter transcript puncta detected in neurites normalized to soma.

**Figure supplement 5.** PRRE motif structural conformations as calculated by RNAfold (*Lorenz et al., 2011*).

*Figure 5 continued on next page*

*Figure 5 continued*

**Figure supplement 6.** Transcript abundance as calculated by smFISH of 3′ PRRE reporter transcripts in LARP1 knockout C2bbe1 cells that have been rescued with either LARP1 or GFP.

**Figure supplement 7.** 3′ PRRE reporter localization in LARP1 knockout C2bbe1 cells that have been rescued with either LARP1 or GFP.

**Figure supplement 8.** Transcript abundance of reporter transcripts as calculated by smFISH of 3′ PRRE reporter transcripts in the soma and neurite compartments of CAD cells that have been treated with either siRNA against LARP1 or a control, nontargeting siRNA.

**Figure supplement 9.** Number of 3′ PRRE reporter transcript puncta detected in neurites normalized to soma in Larp1 siRNA-treated CAD cells.

migrating cancer cells (*Moissoglu et al., 2020*; *Wang et al., 2017*). The element responsible for the localization of mouse *Net1* RNA to neurites has recently been identified as a GA-rich region in the transcript's 3′ UTR, and when it is deleted or targeted with morpholinos, the transcript is no longer localized (*Arora et al., 2022a*; *Chrisafis et al., 2020*).

To ask if the same RNA sequences in *Net1* RNA that drive RNA localization in mouse neuronal cells are also active in human epithelial cells, we created reporter transcripts that contained 3′ UTR sequences derived from mouse *Net1*. These reporter constructs either contained the full length *Net1* 3′ UTR, just the identified GA-rich localization element (LE) or the entire *Net1* 3′ UTR lacking the LE (ΔLE) (*Figure 6A*). As before, the reporter constructs were integrated into the genomes of C2bbe1 cells, inducibly expressed with doxycycline, and the localization of the resulting transcripts was visualized using smFISH. We found that the full-length *Net1* 3′ UTR was sufficient to strongly drive RNAs to the basal pole of epithelial cells. The LE displayed significant yet weaker RNA localization activity, and the 3′ UTR lacking the localization element (ΔLE) displayed no basal localization activity (*Figure 6B and C*, *Figure 6—figure supplement 1*). These results indicate that the GA-rich LE within the mouse *Net1* 3′ UTR is both necessary and sufficient to drive RNA to the basal pole of human enterocytes.

We then assayed the same reporter constructs in mouse neuronal CAD cells using smFISH. As in the human epithelial cells, we found that the full-length *Net1* 3′ UTR strongly drove transcripts to neurites, the LE had significant but reduced activity, and the ΔLE 3′ UTR displayed no activity (*Figure 6D and E*). These results were due to a neurite-specific accumulation of reporter transcripts and not an overall difference in transcript expression (*Figure 6—figure supplement 2*). The GA-rich sequence element within the 3′ UTR of mouse *Net1* therefore represents another sequence that regulates RNA localization across cell types, species, and cellular morphologies.

## Enterocytes and neurons have similar localized transcriptomes

If similar mechanisms were transporting RNAs to the basal pole of epithelial cells and the neurites of neuronal cells, we reasoned that these two subcellular compartments should be enriched for similar RNAs. To test this, we compared LRz and AB values from the neuronal fractionation and epithelial Halo-seq datasets, respectively, transcriptome-wide (*Figure 6F*). Somewhat surprisingly, we found a reasonably strong and highly significant correlation between these values (R=–0.28, p<2.2e-16). The negative sign of this correlation indicates that neurites and the basal pole of epithelial cells are enriched for similar transcripts. RP mRNAs, highlighted in red, exemplify this relationship (R=–0.46, p=1.5e-11). These correlations are even more striking when considering the fact that the LRz and AB values are derived from experiments that differ greatly in their approaches (mechanical subcellular fractionation vs. proximity labeling), species (mouse vs. human), and cell types (neurons vs. intestinal epithelial cells). These very different cellular compartments are therefore spatial equivalents, at least in terms of RNA localization, and their RNA contents are likely governed by the same regulatory mechanisms.

The genes that fall in the fourth quadrant of *Figure 6F* are localized basally in enterocytes and to the neurites of neurons. To interrogate the identity of these genes we used GO enrichment analysis (*Alexa, 2010*). The most enriched GO terms were 'ribosome' and 'mitochondrion' (*Figure 6—figure supplement 3*). We have extensively detailed that RP mRNAs are enriched both in neurites and the basal compartment of epithelial cells (*Figure 3A and I*, *Figure 3—figure supplement 1*). We have similarly found that mitochondria are enriched toward the basal pole of epithelial cells (*Figure 2—figure supplement 8*), and mitochondria have previously been reported to be found in neuronal

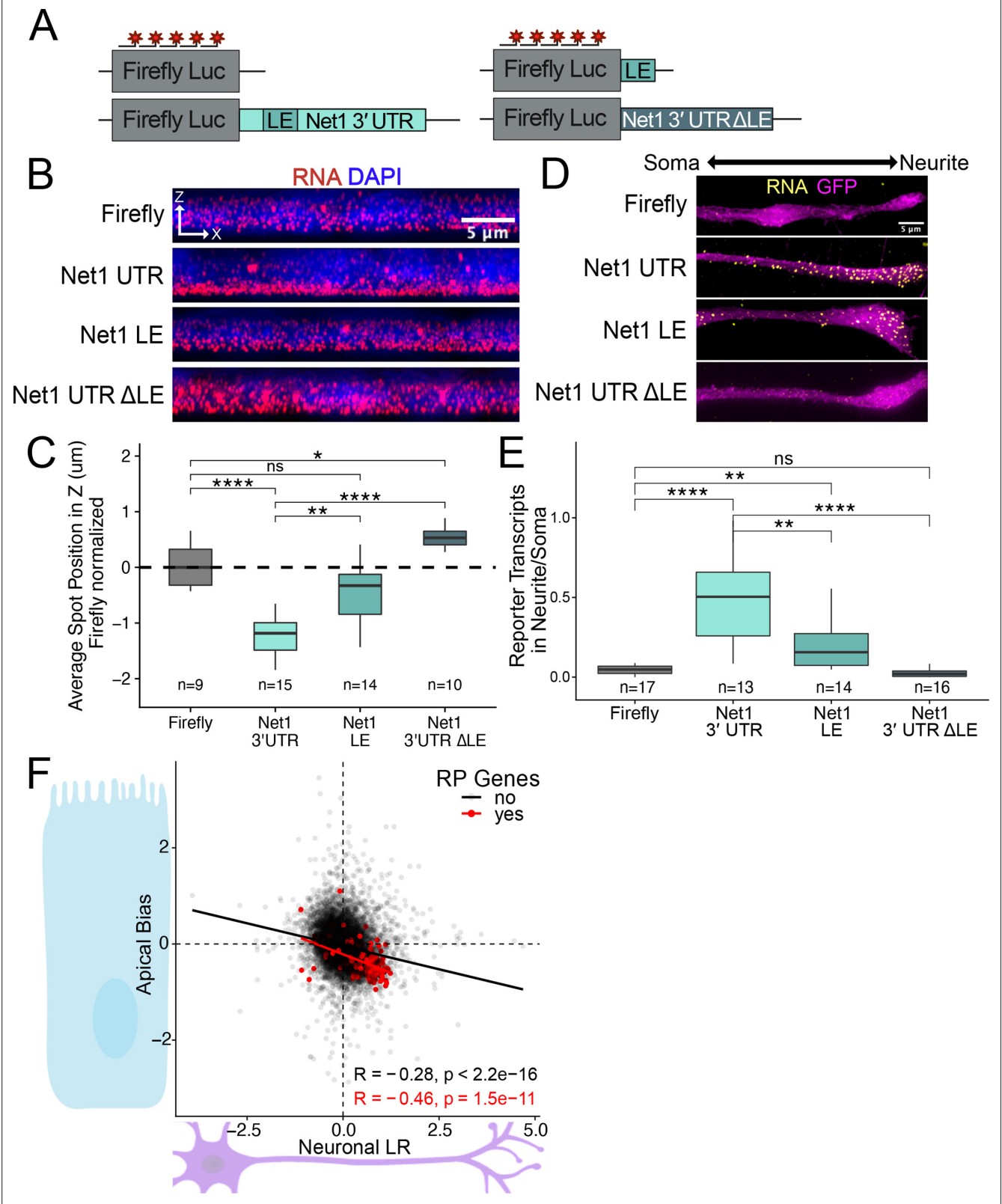

**Figure 6.** RNA localization mechanisms beyond PRRE motifs are conserved across species and cell morphology. (**A**) Schematic of Net1 reporter constructs used to investigate necessity and sufficiency of Net1's localization element (LE). (**B**) Representative images of smFISH puncta for each Net1 reporter construct in C2bbe1 monolayers. RNA signal shown in red while DAPI marks the nuclei in blue. Images are a max projection through the XZ axis of roughly 50 cells. (**C**) smFISH puncta position in Z of the indicated reporter RNAs normalized to median untagged Firefly luciferase transcript position.

*Figure 6 continued on next page*

*Figure 6 continued*

p-Values were calculated using a Wilcoxon rank-sum test. Values represent the average value per cell. Numbers of cells interrogated for each construct are indicated. (**D**) Representative images of smFISH puncta for Net1 constructs in CAD cells. Images are a max projection through the XY axis of a single neurite positioned with the soma to the left. RNA signal shown in yellow while cell outlines marked by GFP signal shown as magenta. (**E**) Number of reporter RNA smFISH puncta in neurites normalized to soma. p-Values were calculated using a Wilcoxon rank-sum test. Numbers of cells interrogated for each construct are indicated. (**F**) Direct comparison of the Halo-seq-derived Apical Bias values in human C2bbe1 monolayers and mechanical fractionation-derived neuronal Localization Ratio (LR) values in neuronal cells for all genes. Ribosomal protein mRNAs are highlighted in red. Correlation and p-values calculated using a Spearman's correlation test. ns (not significant) represents p>0.05, * p<0.05, ** p<0.01, *** p<0.001 and **** represents p<0.0001.

The online version of this article includes the following figure supplement(s) for figure 6:

**Figure supplement 1.** Transcript abundance of reporter transcripts in C2bbe1 monolayers as calculated by smFISH.

**Figure supplement 2.** Transcript abundance in CAD cell neurites and soma as calculated by smFISH.

**Figure supplement 3.** Enriched gene ontology terms derived from RNAs identified as localized to the basal pole of enterocytes and to neurites of neurons.

**Figure supplement 4.** Localization of transcripts encoding electron transport chain (ETC) associated proteins in apicobasal and neuronal localization datasets.

projections (*Mandal and Drerup, 2019*). Given that many nuclear-encoded RNAs that encode mito-chondrial proteins are translated on the surface of the mitochondria, it may be expected that these RNAs are also neurite- and basal-enriched. Consistent with this idea, we found that RNAs that encode components of the electron transport chain (ETC) were both neurite- and basal-enriched according to our sequencing data sets (*Figure 6—figure supplement 4*). This suggests that similarities in RNA localization between these two cell types can be driven by similar organelle localization.

## RNA localization to neurites and the basal pole of epithelial cells is dependent on kinesin activity

Despite drastically different morphologies, intestinal epithelial cells and neurons are organized using similar underlying microtubule structures. These polar molecules are organized in such a way that their plus ends are oriented basally in intestinal epithelial cells and out to neurite tips in neurons (*Müsch, 2004*; *Sugioka and Sawa, 2012*; *Yogev et al., 2016*; *Figure 7A*). Because of the similar microtubule architecture in both cell types and the fact that RNAs are known to be trafficked along microtubules (*Cross et al., 2021*; *Gagnon and Mowry, 2011*), we hypothesized that RNAs were being trafficked to neurites and the basal pole of epithelial cells through the action of kinesin motor proteins. To test this hypothesis, we monitored the localization of our reporter RNAs following treatment of cells with the kinesin-1 inhibitor kinesore (*Randall et al., 2017*).

Cells expressing the full length *Net1* 3' UTR reporter construct were treated for four hours with kinesore or ciliobrevin A, a dynein-1 inhibitor. In C2bbe1 monolayers, we found that kinesin inhibition dramatically reduced the basal localization of reporter RNAs containing the *Net1* 3' UTR (*Figure 7B and C*), consistent with the hypothesis that these RNAs are trafficked along microtubules in epithelial cells. This effect was specific to the *Net1* 3' UTR as the localization of reporter transcripts lacking the 3' UTR was unaffected by kinesin inhibition (*Figure 7—figure supplement 1*). Interestingly, we found that inhibition of dynein strengthens basal localization of Net1 3' UTR transcripts. Kinesin and dynein activity are often closely linked, pulling the same cargoes in different directions. When the activity of one motor is inhibited, it can therefore increase the apparent activity of the other (*Soppina et al., 2009*). In neuronal cells, kinesin inhibition again significantly reduced the neurite localization of the *Net1* 3' UTR reporter RNAs, again suggesting they are transported along microtubules (*Figure 7D and E*). We did not observe any changes in the localization of our control reporter transcript, Firefly luciferase, in response to drug treatments (*Figure 7—figure supplements 1 and 2*). Additionally, as before, changes in RNA localization were driven by changes in RNA content of the neurite and not any gross changes in expression (*Figure 7—figure supplement 3*).

We found very similar results with our PRRE reporter transcripts when inhibiting motor proteins. In C2bbe1 monolayers, kinesin inhibition removed the basal localization of both RPL7 and RPS28 PRRE reporter constructs (*Figure 7F*, *Figure 7—figure supplement 4*). Kinesin-mediated transport along microtubules was also required for the neurite localization of PRRE-containing transcripts (*Figure 7G*).

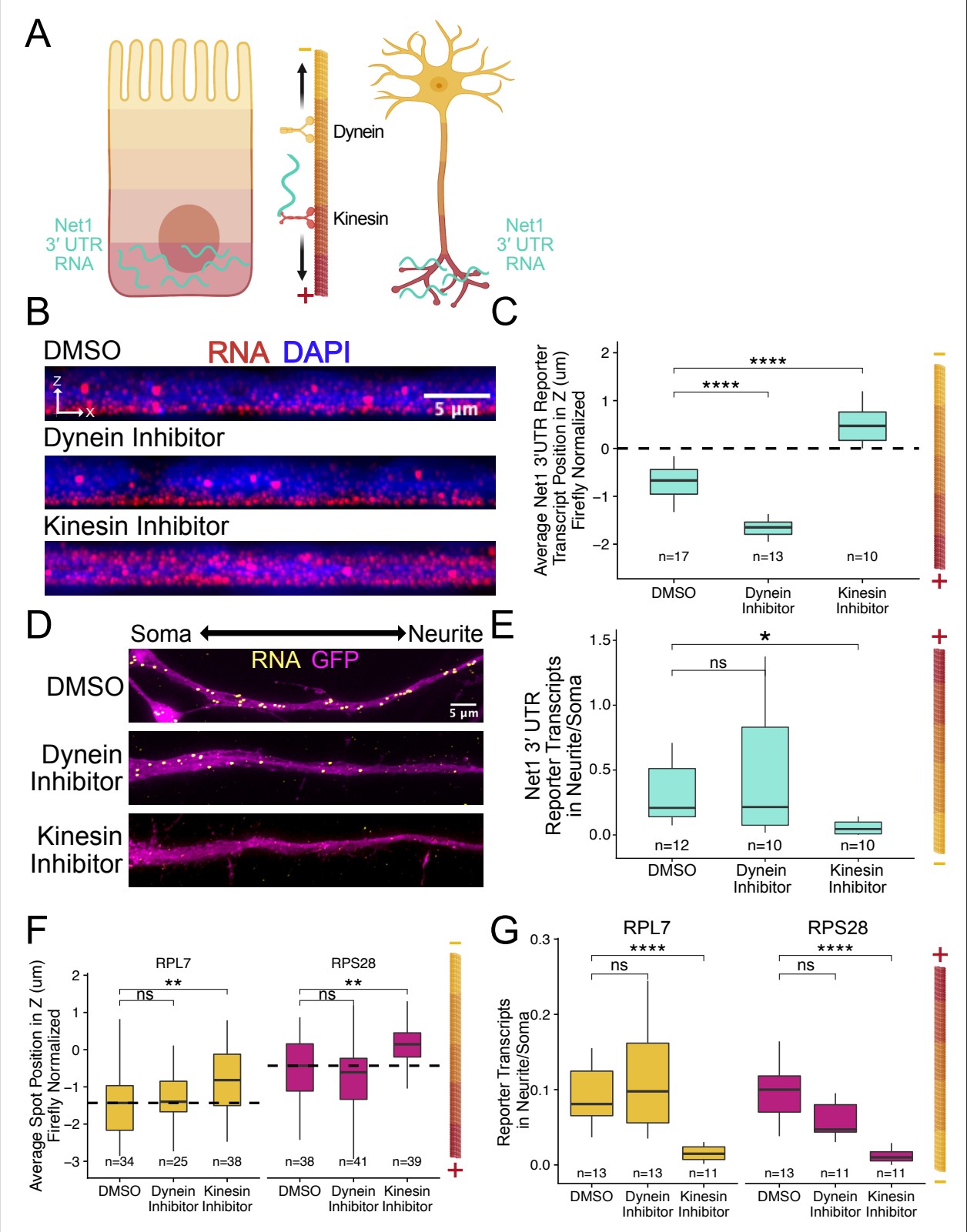

**Figure 7.** RNA localization to neurites and the basal pole of epithelial cells is dependent on kinesin. (**A**) Schematic of microtubule organization in both enterocytes and neurons where the plus ends are oriented basally and out to neurites. Net1 3′ UTR Reporter constructs are teal, hypothesized to reach their destination via kinesin transport. (**B**) Representative images of Net1 3′ UTR reporter transcript localization with each inhibitor treatment or DMSO control. RNA signal shown in red while DAPI marks the nuclei in blue. Images are a max projection through the XZ axis of roughly 30 cells. (**C**) smFISH

*Figure 7 continued on next page*

*Figure 7 continued*

puncta position in Z of Firefly luciferase-Net1 3' UTR constructs normalized to median untagged Firefly luciferase transcript position in Z. p-Values were calculated using a Wilcoxon rank-sum test. Values represent the average value per cell. (D) Representative images of smFISH for Net1 3' UTR transcripts in CAD cells. Images are a max projection through the XY axis of a single neurite positioned with the soma to the left. RNA signal shown in yellow while cell outlines marked by GFP signal shown as magenta. (E) Number of reporter RNA smFISH puncta in neurites normalized to soma. p-Values were calculated using a Wilcoxon rank-sum test. (F) smFISH puncta position in Z of RPL7 and RPS28 PRRE constructs normalized to median untagged Firefly luciferase transcript position in Z. p-Values were calculated using a Wilcoxon rank-sum test. Values represent the average value per cell. (G) Number of reporter RNA smFISH puncta in neurites normalized to soma. p-Values were calculated using a Wilcoxon rank-sum test. ns (not significant) represents p>0.05, * p<0.05, ** p<0.01, *** p<0.001 and **** represents p<0.0001. For panels C, E, F, and G, numbers of cells interrogated for each construct are indicated.

The online version of this article includes the following figure supplement(s) for figure 7:

**Figure supplement 1.** Number of Firefly reporter transcript puncta detected in neurites normalized to soma with different drug treatments in CAD cells.

**Figure supplement 2.** Transcript abundance of reporter transcripts in C2bbe1 monolayers treated with the indicated drugs as calculated by smFISH.

**Figure supplement 3.** Firefly luciferase reporter transcript localization with different drug treatments normalized to median transcript position in DMSO treatment.

**Figure supplement 4.** Abundance of Net1 3' UTR reporter transcripts as assayed by smFISH in the soma and neurite compartments of CAD cells treated with the indicated drugs.

**Figure supplement 5.** Abundance of PRRE-containing reporter transcripts as assayed by smFISH in C2bbe1 monolayers treated with the indicated drugs.

**Figure supplement 6.** Abundance of PRRE-containing reporter transcripts as assayed by smFISH in the soma and neurite compartments of CAD cells treated with the indicated drugs.

Once again, changes in RNA abundance were neurite-specific and were not observed cell-wide (*Figure 7—figure supplement 5*).

Given that the plus ends of microtubules are associated with both neurites and the basal pole of epithelial cells and that kinesin inhibition reduced RNA localization to both compartments, we conclude that the RNA contents of both locations are determined at least in part through microtubule plus-end directed trafficking mechanisms.

## Discussion

For the vast majority of RNAs that display a particular subcellular localization, the RNA elements that regulate this localization are unknown (*Engel et al., 2020*). For those that are known, their activity has generally only been studied in a single cell type (e.g. neurons or *Drosophila* embryos). Because the subcellular location of an RNA is often intricately linked with the morphology of the cell, it has been difficult to predict how RNA localization elements that are active in one cell type would behave in another. In this study, we have demonstrated that two RNA localization elements, PRRE motifs and a GA-rich element in the 3' UTR of the mouse *Net1* RNA, drive RNAs to the projections of mouse neuronal cells and the basal pole of human epithelial cells. Further, transcriptome-wide analysis of these subcellular compartments revealed that their RNA contents are similar, suggesting that the same mechanisms are populating their localized transcriptomes.

RP mRNAs have been previously identified as enriched in neurites (*Taliaferro et al., 2016*) and the basal pole of epithelial cells (*Moor et al., 2017*). However, the mechanisms that direct RP mRNAs to those locations have been unknown. We found that PRRE motifs regulate the trafficking of RNAs to both of those locations and do so in a manner that requires the RBP LARP1. PRRE motifs were also previously found to regulate RNA localization to the protrusions of migrating cells (*Dermit et al., 2020*). However, in these cells, RP mRNA localization required another La-family RBP, LARP6. Larp6 was not expressed in our mouse neuronal model, and LARP6 was only lowly expressed in our human epithelial model. Thus, it is possible that LARP1 and LARP6 have redundancy in RNA localization activities and can complement each other in this regard if expressed in the same cell.

Interactions between LARP1 and PRRE motifs have been well-studied (*Al-Ashtal et al., 2021*; *Lahr et al., 2015*). Generally, these interactions have been characterized in the context of LARP1's ability to regulate the translational status of PRRE-containing RNAs, often through a model in which LARP1's interaction with the 5' cap inhibits translation initiation (*Lahr et al., 2017*). Importantly, it has been demonstrated that LARP1 contains at least three RNA binding regions, a DM15 region that recognizes

the 5′ mRNA cap, an RNA recognition motif, and a La motif that recognizes the PRRE. RNA binding is tightest when both of these binding modes (i.e. to the 5′ cap and the PRRE motif) are engaged (*Lahr et al., 2017*). These models have further proposed that regulation by LARP1 requires the 5′ cap and pyrimidine-rich motif be *immediately* adjacent to each other. We challenge the necessity of this adjacency for two reasons.

First, although the *presence* of PRREs within 5′ UTRs was clearly connected with RNA localization, we found no relationship between the *position* of the PRREs within 5′ UTRs and their ability to regulate RNA localization. Conversely, PRREs in 3′ UTRs were unable to regulate RNA localization. Together, this suggests that general proximity to the 5′ cap may be important although immediate adjacency may not be required. Second, multiple datasets interrogating transcription start sites using CAGE and related techniques have found that for most mammalian genes, including RP-mRNAs, transcription is not initiated at the same nucleotide every time (*Noguchi et al., 2017*). Instead, transcription begins somewhere within a window that is often several nucleotides wide. In this case, only some fraction of RP-mRNA transcripts would have their 5′ cap and pyrimidine-rich motifs located immediately adjacent to each other.

Intriguingly, the depletion of Larp1 in mouse neuronal cells and the knockout of LARP1 in human epithelial cells led to noticeable phenotypes. In neuronal cells, Larp1 depletion led to cells producing fewer and shorter neurites. LARP1 knockout epithelial cells had a noticeable growth defect and had considerably longer doubling times. It is tempting to ascribe at least parts to these phenotypes to the mislocalization of RP mRNAs, especially since RP mRNAs are some of the most highly expressed mRNAs in the cell. However, given that LARP1 participates in multiple facets of post-transcriptional gene regulation (*Berman et al., 2021*), we cannot directly make this conclusion.

The conserved localization of RP mRNAs as a class across multiple species and cell types suggests a repeated need for the spatial control of ribosomal protein synthesis. The translation of RP mRNAs could greatly impact the translational output of an entire cell as more ribosomal proteins allows for a greater number of functional ribosomes (*Fonseca et al., 2018*). Still, given that ribosome assembly canonically occurs in the nucleolus, it is puzzling as to why RNAs encoding ribosomal subunits would often be kept so far away from the nucleus. One idea is that locally synthesized ribosomal proteins can directly join and remodel, or perhaps repair, nearby ribosomes, promoting further local control of translation (*Shigeoka et al., 2019*).

We also found that a GA-rich region in the 3′ UTR of the mouse *Net1* RNA is necessary and sufficient to drive RNAs both to the neurites of mouse neuronal cells and the basal pole of human epithelial cells. Interestingly, a GA-rich region in the 3′ UTR of human *NET1* RNA was also identified as necessary for the localization of the transcript to the leading edges of migrating cells (*Chrisafis et al., 2020*) dependent on transport by KIF1C (*Pichon et al., 2021*).

Overall, these findings point to the existence of an RNA localization 'code' that operates in multiple cell types and is conserved across species. In the mechanisms illuminated in this study, we propose that the RNA localization codes we identified are not 'neurite localization' signals or 'basal localization' signals per se. Rather, they are signals that specify that an RNA should be trafficked to the plus end of microtubules, wherever that may be in a given cell type. This opens up the possibility of predicting the localization of an RNA in a given cell type if its localization in another cell type is known, greatly increasing our understanding of mechanisms underlying RNA trafficking.

## Methods

**Key resources table**

| Reagent type (species) or resource | Designation | Source or reference | Identifiers | Additional information |
|---|---|---|---|---|
| Gene (*Homo sapiens*) | P65 | hg38 | ENSG00000173039 | |
| Gene (*Homo sapiens*) | PODXL | hg38 | ENSG00000128567 | |
| Gene (*Homo sapiens*) | ATP1A1 | hg38 | ENSG00000163399 | |

*Continued on next page*

*Continued*

| Reagent type (species) or resource | Designation | Source or reference | Identifiers | Additional information |
|---|---|---|---|---|
| Gene (*Homo sapiens*, *Mus musculus*) | RPL7 | hg38, mm10 | ENSG00000147604, ENSMUSG00000043716 | |
| Gene (*Homo sapiens*, *Mus musculus*) | RPS28 | hg38, mm10 | ENSG00000233927, ENSMUSG00000067288 | |
| Gene (*Homo sapiens*, *Mus musculus*) | LARP1 | hg38, mm10 | ENSG00000155506, ENSMUSG00000037331 | |
| Gene (*Homo sapiens*, *Mus musculus*) | NET1 | hg38, mm10 | ENSG00000173848, ENSMUSG00000021215 | |
| Sequence-based reagent | LARP1 guide RNA exon2 | Synthego | GCUGUUCCUAAACAGCGCAA | Used to create human LARP1-null cells |
| Sequence-based reagent | LARP1 guide RNA exon19 | Synthego | CAACCCCCUACACCACCCAC | Used to create human LARP1-null cells |
| Sequence-based reagent | AAVS1 guide RNA | Synthego | UAGUGGCCCCACUGUGGGGU | Used to create human loxP cells |
| Sequence-based reagent | Larp1 siRNA | IDT | mm.RI.Larp1.13.2 | Used as equimolar mix |
| Sequence-based reagent | Larp1 siRNA | IDT | mm.RI.Larp1.13.3 | Used as equimolar mix |
| Sequence-based reagent | SSB siRNA | IDT | mm.RI.Larp1.13.1 | Used alone |
| Sequence-based reagent | Larp4 siRNA | IDT | mm.RI.Larp1.13.1 | Used alone |
| Sequence-based reagent | LARP7 siRNA | IDT | mm.RI.Larp1.13.1 | Used alone |
| Sequence-based reagent | FLAP_Y | IDT | /5Cy3/AA TGC ATG TCG ACG AGG TCC GAG TGT AA/3Cy3Sp/ | hybridized to smiFISH probes |
| Sequence-based reagent | RPL7 5'TOP | IDT | CCTCTTTTTCCGGCTGGAACC | used for cloning into reporter constructs |
| Sequence-based reagent | RPL7 mutant 5'TOP | IDT | AAGAGGGGGAAGGAGGGAAAA | used for cloning into reporter constructs |
| Sequence-based reagent | RPS28 5'TOP | IDT | ACTCCTCTCCGCCAGACCGCCGCCGCGCCGCCATC | used for cloning into reporter constructs |
| Sequence-based reagent | RPS28 mutant 5'TOP | IDT | AAGAAGAGAAGAAAGAAAGAAGAAGAGAAGAAAGA | used for cloning into reporter constructs |
| Cell line (*Mus musculus*) | CAD | Sigma | 08100805-1VL, RRID:CVCL_0199 | |
| Cell line (*Mus musculus*) | CAD/loxP | **Khandelia et al., 2011** | | Contains single integration of loxP cassette |
| Cell line (*Homo sapiens*) | C2bbe1 | ATCC | CRL-2102 | |
| Cell line (*Homo sapiens*) | C2bbe1/loxP | **Khandelia et al., 2011** | | Contains single integration of loxP cassette |

*Continued on next page*

*Continued*

| Reagent type (species) or resource | Designation | Source or reference | Identifiers | Additional information |
|---|---|---|---|---|
| Cell line (*Homo sapiens*) | HCA-7 | ECACC | 6061902 | |
| Cell line (*Homo sapiens*) | HCA-7/loxP | ***Khandelia et al., 2011*** | | Contains single integration of loxP cassette |
| Cell line (*Canis lupus*) | MDCK | ATCC | CRL-2935 | |
| Cell line (*Canis lupus*) | MDCK/loxP | ***Khandelia et al., 2011*** | | Contains random integration of loxP cassette |
| Transfected construct (*Homo sapiens*) | PODXL cDNA | Horizon | MHS6278-202858197 | used for cloning into Halo plasmids |
| Transfected construct (*Homo sapiens*) | ATP1A1 cDNA | Horizon | MHS6278-202759485 | used for cloning into Halo plasmids |
| Transfected construct (*Homo sapiens*) | P65 cDNA | R.Spitale, UC Irvine | | used for cloning into Halo plasmids |
| Transfected construct (*Homo sapiens*) | LARP1 cDNA | Horizon | MHS6278-202827213 | used for cloning into Halo plasmids |
| Antibody | mouse monoclonal anti LARP1 | Santa Cruz | sc-515873 | 1:100 dilution for immunoblotting |
| Antibody | mouse monoclonal anti ACTB | Sigma | A5441 | 1:5000 for immunoblotting |
| Antibody | mouse monoclonal anti H3 | Abcam | 10799 | 1:10000 for immunoblotting |
| Antibody | rabbit polyclonal anti TOM20 | ProteinTech | 11802–1-AP | 1:250 for immunofluorescence |
| Antibody | rabbit polyclonal anti EZRIN | Cell Signaling Technology | 3145 S | 1:250 for immunofluorescence |
| Antibody | mouse monoclonal anti NaKATPase | DSHB | A5-s | 1:50 for immunofluorescence |
| Sequence-based reagent | smFISH probes against Firefly luciferase | BioSearch | VSMF-1006–5 | |
| Commercial assay or kit | Zymo Quick-RNA Microprep kit | Zymo Research | R1050 | |
| Commercial assay or kit | Zymo Quick-RNA Miniprep kit | Zymo Research | R1055 | |
| Commercial assay or kit | Roche RNA HyperPrep Kit with HMR depletion | Roche | | |
| Other | Cell culture inserts for differentiation | Corning | 353090 | C2bbe1 are differentiation on 0.4 uM transwell inserts |
| Other | Kinesore | Sigma-Aldrich | SML2361 | Kinesin-1 Inhibitor |
| Other | Ciliobrevin-A | Selleckchem | 302803-72-1 | Dynein Inhibitor |

*Continued on next page*

*Continued*

| Reagent type (species) or resource | Designation | Source or reference | Identifiers | | Additional information |
|---|---|---|---|---|---|
| Other | MitoView Green | Biotium | 70,054 T | | 100 ng/mL for immunofluorescence imaging of mitochondria |
| Other | Halo-Dibromofluoroscein | R.Spitale, UC Irvine | | | ROS producing small molecule for Halo-seq labeling |

## Cell maintenance, differentiation

C2bbe1 cells (CRL-2102) were maintained in DMEM with 110 mg/L Sodium Pyruvate (Thermo 11995065) supplemented with 10% FBS (Atlas Biologicals F-0500-D), penicillin-streptomycin (Thermo 15140122) and 10 µg/mL human transferrin (Sigma-Aldrich T8158). HCA-7 cells were maintained in a 1:1 mix of DMEM and F12 (Thermo 11320033) supplemented with 10% FBS (Atlas Biologicals F-0500-D) and penicillin-streptomycin (Thermo 15140122). MDCK cells were maintained in DMEM (Thermo 11965092) supplemented with 10% Equafetal (Atlas Biologicals EF-0500-A) and penicillin-streptomycin (Thermo 15140122).

C2bbe1 LoxP cells were maintained in 20 µg/mL blasticidin (A.G. Scientific B-1247-SOL) while HCA-7 and MDCK lox cells were maintained in 10 µg/mL blasticidin. Following *cre*-mediated recombination with a puromycin resistance plasmid, all reporter-expressing lines were maintained in 5 µg/mL puromycin (Cayman Chemical Company 13884).

C2bbe1, HCA-7 and MDCK cells were allowed to differentiate for 7 days after plating at 100% confluency in the presence of 2 µg/mL doxycycline (Fisher Scientific AAJ6042203) when inducing transgene and/or reporter transcript expression. Differentiation can occur in dishes, plates or on 0.4 µm transwell inserts (Corning 353090). Media was changed every 3–4 days.

CAD cells were maintained in a 1:1 mix of DMEM and F12 (Thermo 11320033) supplemented with 10% Equafetal (Atlas Biologicals EF-0500-A) and penicillin-streptomycin (Thermo 15140122). CAD lox cells were received as a gift from Eugene Makeyev (*Khandelia et al., 2011*) and maintained in 5 µg/mL blasticidin (A.G. Scientific B-1247-SOL).

CAD cells were grown in full growth media and induced with 2 µg/mL doxycycline for 48 hr when inducing expression of transgenic constructs. Following doxycycline induction, media was replaced with serum-free media for 48 hr which induces neurite outgrowth. Doxycycline induction was continued through the differentiation period when inducing construct expression. Reporter-expressing CAD lines were maintained in 5 µg/mL puromycin (Cayman Chemical Company 13884).

All cell lines were tested for mycoplasma and found to be negative.

## CRISPR/Cas9 modifications

A loxP-flanked blasticidin resistance cassette was integrated into the AAVS1 safe harbor of C2bbe1 and HCA-7 cells via CRISPR/Cas9. RNPs from Synthego with AAVS1 targeting sgRNA (UAGUGGC-CCCACUGUGGGGU) were electroporated with the Neon electroporation system (C2bbe1:1600 V 10ms and 3 pulses, HCA-7: 1100 V 20ms 2 pulses). A donor plasmid containing the loxP cassette with homology to the AAVS1 safe harbor was co-electroporated with the RNPs. Following electroporation, cells were incubated at 37 °C for 48 hr before selection with blasticidin (20 µg/mL for C2bbe1, 10 µg/mL for HCA-7) until untransfected negative control cells had died and the newly selected loxP line exhibited normal growth patterns.

Integration of the loxP cassette into the AAVS1 safe harbor was validated by PCR amplifying the junctions of the AAVS1 site upstream and downstream the inserted cassette. Primers were designed such that PCR products are only created when the LoxP cassette is inserted as expected. The parental line was included as a negative control and did not produce junction PCR products.

MDCK loxP cells were created through random integration using the same electroporation (1600 V 10ms 3 pulses) strategy and donor plasmid for human lines. CRISPR/Cas9 RNPs were omitted from this electropporation. Selection of blasticidin resistant clones contained loxP cassettes in undetermined locations.

For the LARP1 knockout line, C2bbe1 loxP cell lines were treated with two sgRNAs from Synthego that targeted exon 2 (GCUGUUCCUAAACAGCGCAA) and exon 19 (GCUGUUCCUAAACAGCGCAA) of LARP1. CRISPR Cas9 RNPs and a GFP plasmid were co-electroporated using the Neon electroporation system (1600V 10ms and 3 pulses) and incubated at 37 °C for 48 hr. GFP positive cells were sorted to single cells using a flow cytometer (MoFlo XDP100) and allowed to expand for 2–4 weeks. Clones were screened for gene deletion by PCR as well as LARP1 mRNA and protein depletion determined by qPCR and western blot analysis with a monoclonal LARP1 antibody (Santa Cruz sc-515873). Lesions at the locus of cutting were identified by PCR using genomic DNA to amplify the locus followed by Sanger sequencing and Synthego ICE analysis.

## Plasmid construction

All plasmids are derivatives of pRD-RIPE (*Khandelia et al., 2011*). All Halo-fusion proteins were cloned into the location of the GFP ORF by excising GFP with AgeI and BstXI and then PCR amplifying the cDNA of the Halo fusion protein of interest with overhanging primer handles that provide 20 bp of homology to the plasmid backbone. Some Halo fusion cDNAs were synthesized by Twist Bioscience. PODXL (Horizon, MHS6278-202858197), ATP1A1 (Horizon, MHS6278-202759485), and LARP1 (Horizon, MHS6278-202827213) cDNAs were obtained from Horizon Discovery. Infusion cloning (TakaraBio) was used to ligate PCR amplified inserts into plasmid backbones. Constructs were confirmed by restriction digest assays and DNA sequencing.

A variant of pRD-RIPE with a bi-directional promoter driving both Firefly and Renilla luciferases was inserted into pRD-RIPE creating pRD-RIPE-BiTET-FR. Reporter constructs were created by adding additional sequence to cut sites in the 5' (AscI) or 3' (PmeI) end of the Firefly luciferase ORF. Because PRRE motifs are short, DNA inserts were created by annealing two oligos and cloning via sticky end cloning with T4 DNA Ligase (NEB M0202S) into the linearized pRD-RIPE-BiTET-FR such that inserted motifs were 90nt downstream of the canonical CMV promoter transcription start site (*Isomura et al., 2008*).

## Cre recombinase-mediated cassette switching

All loxP-containing cell lines followed the same protocol for cassette switching. Cells were plated in 12-well plates at $1.0–1.5\times10^5$ cells per well in full growth media 12–18 hr before transfection. Cells were then co-transfected with 500 ng donor cassette plasmid with 1% (wt/wt) of a Cre-encoding plasmid (pBT140, addgene #27493). Donor plasmids with doxycycline inducible transgenes also contained a constitutively active copy of the tetracycline transactivator. Plasmids were transfected using Lipofectamine LTX reagent (Thermo Fisher Scientific 5338030) and Opti-MEM (Thermo Fisher Scientific 31985070) following the manufacturer's protocol. Following 24 hr with transfection reagents, the medium was changed and the incubation continued for another 24 hr before addition of 1–5 µg/mL puromycin (VWR 97064–280). Incubation with puromycin continued for several days until untransfected control wells were depleted of cells. The puro-resistant colonies were expanded in a full growth medium supplemented with 5 µg/mL puromycin.

## Halo SDS-PAGE gel stain

Halo-expressing cells were grown in the presence or absence of 2 µg/mL doxycycline for 48 hr before lysate collection. Following a PBS wash, cells were incubated with 25 nM Halo ligand Janelia Fluor 646 (Promega GA1120) for 10 min. Cells were washed three times with PBS. Lysates from 1.0 to $1.5\times10^6$ cells were scraped into 50–75 µL RIPA lysis buffer (50 mM Tris-HCl, pH 7.0, 150 mM NaCl, 0.1% (w/v) SDS, 0.5% sodium deoxycholate, 1.0% Triton X, 5 mM EDTA). Lysates were chilled on ice for 15 min before sonication for 30 s. 15 µL of lysates were mixed with sample buffer (Invitrogen NP0008) before being incubated at 100 °C for 5–10 min. Samples were loaded into 4–12% Bis-Tris 1 mm SDS protein gel (Thermo Fisher Scientific NP0323BOX) with protein ladder (Gel Company FPL-008). Gels were imaged with a Sapphire molecular imager (Azure Biosystems) set to collect fluorescent Cy5 signal. Following fluorescent imaging the gel was stained with 1% Coomassie (VWR 95043–420) for 20 min, de-stained (40% Methanol, 10% Acetic Acid) 1–16 hr, and imaged with the visible setting.

## Halo imaging in fixed cells

Halo fusion proteins were visualized in cells by staining with fluorescent Halo ligand (Promega GA1110). Cells were seeded on polyD lysine-coated coverslips (neuVitro) in full growth media supplemented with 2 µg/mL doxycycline to induce Halo transgene expression for 48 hr. Following a PBS wash, cells were incubated with Halo ligand Janelia Fluor 549 (Promega GA1110) at 25 nM for 30 min at room temperature. Following PBS wash, cells were fixed with 3.7% formaldehyde for 20 min at room temperature. Cell nuclei were stained with 100 ng/mL DAPI (Sigma-Aldrich D9542) for 15 min, and then cells were washed again with PBS. Coverslips were mounted with fluoromount G (SouthernBiotech 0100–01) and imaged at ×60 magnification with a Deltavision Elite widefield fluorescence microscope (GE). DAPI was imaged with 5% transmittance with an exposure of 0.3 s. Halo staining was visualized in the TRITC channel with 10% transmittance and an exposure of 0.2 s. Z stacks were collected of approximately 68 images, 0.2 µm apart for a total thickness of 13.6 µm.

## Halo-seq

Halo-seq was performed as previously described (*Engel et al., 2021*; *Lo et al., 2022*). Details regarding each step of Halo-seq are provided below.

## In-cell RNA proximity labeling

Halo transgene-expressing C2bbe1 cells were seeded at high confluency on 0.4 µm PET transwell inserts (Corning 353090). Nine 6-well transwells were pooled for each replicate sample. Monolayers were differentiated for 7 days and Halo fusion proteins were induced with 0.5 µg/mL doxycycline. Following PBS wash, cells were incubated with 1 µM DBF Halo ligand in HBSS (gift from Rob Spitale) at 37 °C for 15 min. Negative control samples lacking DBF were incubated in HBSS only.

Cells were washed with full growth media for 10 min at 37 °C twice. Immediately following this, cells were incubated in 1 mM propargylamine (Sigma-Aldrich P50900) in HBSS for 5 min at 37 °C. Cells were then moved to the green light chamber containing two 100 W LED flood lights (USTELLAR) in an enclosed dark space. Cells were irradiated with green light for 10 min in the room temperature chamber sandwiched between two flood lights. Cells were collected in Trizol (Ambion) and homogenized by passing lysates through a 20 G needle 20 times. Total RNA was then isolated following Trizol's manufacturer instructions.

RNA was DNase treated using DNase I (NEB M0303S) for 30 min at 37 °C. RNA was then recovered using Quick RNA columns (Zymo Research R1055) following the sample clean up protocol and eluted in 100 µL water.

## In vitro biotinylation of RNA via 'Click' chemistry

Approximately 200 µg of total RNA was used in the Click reaction. The Click reaction contained 10 mM Tris pH 7.5, 2 mM biotin picolyl azide (Sigma-Aldrich 900912), 10 mM sodium ascorbate made fresh (Sigma 11140), 2 mM THPTA (Click Chemistry Tools 1010–100), and 100 µM copper sulfate (Fisher Scientific AC197722500). Click reactions were split into 50 µL reaction volumes and then incubated for 30 min in the dark at 25 °C. Click reactions were pooled and cleaned using a Quick RNA Mini kit (Zymo Research R1055) following the sample clean up protocol and eluted in 100 µL water. Negative control samples where cells were not treated with DBF were also biotinylated using this procedure.

## Streptavidin pull down

A total of 100 µg of biotinylated RNA at 1 µg/µL was used as the input for streptavidin pulldowns. Different volumes of streptavidin-coated magnetic beads (Pierce PI88816) were used depending on the amount of labeled RNA by each Halo fusion protein. Halo-P65 required 1 µl beads per 2 µg total RNA while apical-Halo and basal-Halo required 1 µl beads per 4 µg RNA. Prior to mixing with RNA, beads were washed three times in B&W buffer (5 mM Tris pH 7.5, 0.5 mM EDTA, 1 M NaCl, 0.1 Tween 20), two times in solution A (0.1 M NaOH, 50 mM NaCl), and one time in solution B (100 mM NaCl). The beads were resuspended with 100 µg RNA and 50 mM NaCl to a total volume of 100 µL. The pulldown reaction was rotated for 2 hr at 4 °C. The beads were then washed three times for 5 min each in B&W buffer with rotation at room temperature.

RNA was then eluted from the beads by resuspending in 50 µL PBS and 150 µL Trizol and incubated for 10 min at 37 °C. The eluted RNA was recovered from this mixture using a DirectZol micro

RNA columns (Zymo Research R2062) following the manufacturer's instructions and eluted in 10 μL of water. The eluted RNA concentration was measured by Qubit high sensitivity RNA kit (Invitrogen Q32855) and +DBF and -DBF samples were compared.

## RNA dot blots

The efficiency of the biotinylation reaction and the streptavidin pulldown were assayed by RNA dot blots. Hybond-N +membrane (GE) was wet with 2 X SSC then allowed to dry for 15 min. Five μg of biotinylated RNA samples or 1% of streptavidin eluted RNA were spotted on the membrane and allowed to dry for another 30 min. To stain for total RNA, the blot was incubated in 1% methylene blue (VWR 470301–814) for 2 min and destained using deionized water. Next, the membrane was blocked using 5% BSA for 30 min and washed three times in PBST (PBS +0.01% Tween). Biotinylated RNA was then detected using streptavidin-HRP (Abcam ab7403) at a dilution of 1–20,000 in 3% BSA by addition to the membrane with rocking overnight at 4 °C. Membranes were washed three times for 10 min each in PBST at room temperature. Streptavidin-HRP was detected using standard HRP chemiluminescent reagents (Advansta) and visualized using chemiluminescent imaging on a Sapphire molecular imager (Azure Biosystems).

## Imaging of alkynylated molecules in cells

Halo-transgene expressing C2bbe1 cells were plated at high confluency on poly-D-lysine-coated coverslips (neuVitro). Monolayers were differentiated for 7 days with 0.5 μg/mL Doxycycline to induce Halo fusion protein expression. Media was changed every 3–4 days. Cells were treated with DBF, propargylamine, and irradiated with green light exactly as described for Halo-seq. Instead of lysing cells with Trizol to recover RNA, cells were washed with PBS 3 times and fixed and permeabilized with 3.7% formaldehyde and 0.3% Triton-X (VWR 80503–490) in PBS for 30 min at room temperature. Following PBS wash, in situ click reactions were performed by incubating cells with 100 μL Click buffer (100 μM copper sulfate, 2 mM THPTA, 10 mM fresh sodium ascorbate, 10 μM Cy3 picolyl azide (Click Chemistry Tools 1178–1)) for 1 hr at 37 °C in the dark. Two negative controls were included: samples with DBF omitted from the in-cell labeling reaction and cells with Cy3 picolyl azide omitted from the in situ Click reaction. Following the Click reaction, the coverslips were washed three times for 5 min each in wash buffer (0.1% Triton, 1 mg/mL BSA in PBS). Next, coverslips were incubated in wash buffer supplemented with 100 ng/mL DAPI for 30 min at 37 °C and then washed twice with wash buffer for 5 min each. The coverslips were mounted with fluoromount G and imaged using a Deltavision Elite widefield fluorescence microscope (GE). DAPI was imaged with 10% transmittance with an exposure of 0.05 seconds. Cy3 azide was visualized in the TRITC channel with 32% transmittance and an exposure of 0.2 s. Z stacks were collected of approximately 51 images, 0.2 μm apart for a total thickness of 10.2 μm.

## Library preparation and sequencing

rRNA-depleted RNAseq libraries were prepared using an RNA HyperPrep Kit (KAPA / Roche). A total of 100 ng of RNA was input into the procedure and fragmented for 3.5 min at 94 °C. Libraries were amplified using 14 PCR cycles.

Libraries were sequenced using paired end sequencing (2x150 bp) on a NovaSeq high-throughput sequencer (Illumina) at the University of Colorado Genomics Core Resource. Between 20 and 40 million read pairs were sequenced for each sample.

## Computational strategy for HaloSeq

Library adapters were removed using cutadapt 2.1 (*Martin, 2011*). Transcript abundances in RNAseq data were quantified using Salmon 0.11.1 (*Patro et al., 2017*) and a human genome annotation retrieved from GENCODE (https://www.gencodegenes.org/, GENCODE 28). Gene abundances were then calculated from these transcript abundances using tximport (*Soneson et al., 2015*), and genes whose abundance in input and streptavidin-pulldown samples were identified using DESeq2 (*Love et al., 2014*).

## Analysis of pre-mRNA transcripts in RNAseq data

To quantify the relative abundances of pre-mRNA (including introns) and mature mRNA (all introns spliced), a custom fasta file was supplied to Salmon which contained two versions of every transcript,

one with all introns remaining and one with all introns removed. The custom fasta file was generated using this script: https://github.com/rnabioco/rnaroids/blob/master/src/add_primary_transcripts.py (*Riemondy, 2018*). Salmon then assigned reads competitively to these transcripts.

## Immunofluorescence staining in monolayers

C2bbe1 cells were seeded on PDL coated glass coverslips (neuVitro) within 12-well plates at high confluency. Monolayers were allowed to differentiate for 7 days, and media was changed every 3–4 days. Following PBS wash, cells were fixed with 3.7% formaldehyde at room temperature for 15 min. After another PBS wash, cells were permeabilized for 30 min at room temperature with 0.3% Triton-X (VWR 80503–490) in PBS. Primary antibodies with various dilutions (Ezrin, Cell Signaling Technology 3145 S, 1:250; NaK ATPase, DSHB A5-s, 1:50; TOM20, ProteinTech 11802–1-AP, 1:250) were diluted in PBS containing 1% BSA and 0.3% Triton-X for 1 hr at 37 °C. Cells were then rinsed three times with PBS before adding 100 ng/mL DAPI and fluorescent secondary antibody (Cell Signaling Technology 4413 S) diluted 1:500 in PBS containing 1% BSA and 0.3% Triton-X for 1 hr at room temperature. When imaging for mitochondria, 100 ng/mL MitoView Green (Biotium 70,054 T) is added at this point as well. Following PBS wash, cells were mounted onto slides with Fluoromount G (SouthernBiotech 0100–01) and sealed with nail polish. Slides were imaged on a widefield DeltaVision Microscope at 60 X (GE) with consistent laser intensity and exposure times across samples. DAPI was imaged with 10% transmittance with an exposure of 0.25 s. Antibodies are visualized in the TRITC channel with 5–32% transmittance and an exposure of 0.3–0.5 s. Z stacks were collected of approximately 63 images, 0.2 µm apart for a total thickness of 12.6 µm.

## SmiFISH probe design and preparation

To assay RNA localization of endogenous genes, the R script Oligostan (*Tsanov et al., 2016*) was used to design primary probes for each RNA of interest (*Supplementary file 2*). Probes were 26–32 nt in length with 40–60% GC content and repetitive sequences were avoided. Primary probes were designed with excess sequence for hybridization to a fluorescence FLAP sequence.

FLAP Y (TTACACTCGGACCTCGTCGACATGCATT) covalently attached to Cy3 on both ends was used for all smiFISH experiments targeting endogenous RNA. 1 X pmol of Primary probes and 1.2 X pmol FLAP sequences were hybridized in NEB buffer 3 in a thermocycler. The reaction was heated to 85 °C for 3 min, cooled to 65 °C for 3 min then held at 25 °C yielding hybridized smiFISH probes specific to the endogenous RNA target.

## Stellaris probe design and preparation

Customized Stellaris smFISH probes were designed and ordered using the freely available probe designer software by Biosearch Technologies. Probes that were designed across the open reading frame of Firefly luciferase were used to assay localization of Firefly reporter constructs. Probes were resuspended in water to a final concentration of 12.5 µM.

## Cell preparation and hybridization for smFISH experiments

Generally, FISH experiments were performed as previously described (*Arora et al., 2022b*). CAD cells expressing reporter constructs were induced with 2 µg/mL doxycycline for 48 hr. Cells were then plated on PDL coated glass coverslips (neuVitro) within 12-well plates at approximately $2.5 \times 10^4$ cells per well in full growth media. Cells were allowed to attach for 2 hr before changing to serum depleted media supplemented with 2 µg/mL doxycycline for 48 hr.

C2bbe1 cells expressing reporter constructs were seeded on PDL coated glass coverslips at high density and allowed to differentiate into monolayers for 7 days in full growth media supplemented with 2 µg/mL doxycycline. Media was changed every 3–4 days. Cells were washed once with PBS before being fixed with 3.7% formaldehyde (Fisher Scientific) for 10 min at room temperature. Following 2 PBS washes, cells were permeabilized with 70% Ethanol (VWR) at 4 °C for 6–8 hr or at room temperature for 2 hr. Cells were incubated in smFISH wash buffer (2 x SSC with 10% formamide) at room temperature for 5 min. Per coverslip, 2 µL of Stellaris FISH Probes, or hybridized smiFISH probes were added to 200 µL of hybridization Buffer (10% dextran sulfate, 10% formamide in 2 X SSC). Using a homemade hybridization chamber made from an empty tip box with wet paper towels and parafilm, coverslips were incubated cell side down in the hybridization buffer overnight at 37 °C. Coverslips

were washed with wash buffer in fresh 12-well plates cell side up for 30 min at 37 °C in the dark. A total of 100 ng/mL DAPI was diluted in wash buffer and added to the cells in the dark at 37 °C for 30 min. Cells were then washed for 5 min at room temperature with wash buffer.

Coverslips were then mounted onto slides with Fluoromount G and sealed with nail polish. Slides were imaged at 60 X on a widefield DeltaVision Microscope with consistent laser intensity and exposure times across samples. DAPI was imaged with 10% transmittance with an exposure of 0.05 s. FISH probes were visualized in the TRITC channel with 100% transmittance and an exposure of 0.15 s. Z stacks were collected of approximately 68 images, 0.2 μm apart for a total thickness of 13.6 μm.

## smFISH computational analysis

FISH-quant (*Mueller et al., 2013*) was used to identify smFISH puncta within CAD neurites and somas through use of its built-in Gaussian filtering and spot detection. All spot thresholding parameters were set to exclude gross outliers. The detection settings were the same for every soma and neurite imaged within an experiment. Approximately ten cells were analyzed for each sample. The mean number of Firefly luciferase transcripts in neurites compared to soma was calculated by FISH-quant mature mRNA summary files. Cell outlines for somas and neurites were manually drawn within the FISH-quant software. Usually cell outlines were visible in the FISH channel with high brightness. Outlines were drawn for proximal neurites that did not overlap with other cell's neurites, protruded across the field of view, and were easily associated with the correct soma.

Images of epithelial monolayers were analyzed similarly with cell outlines manually drawn and approximated from GFP expression. Despite imaging a monolayer, smFISH spots were only quantified from outlined cells entirely within the field of view with sufficient RNA expression. Instead of using summary files, all detected spot files were used to ensure thresholding included spots from positive controls while excluding spots detected in negative control samples in which the RNA species (e.g. Firefly luciferase) being probed was not expressed.

Firefly luciferase reporter transcript puncta were thresholded until an expected expression of 100–1000 spots per cell was achieved. smFISH puncta counts of different transcripts were expected to approximately reflect relative expression values calculated from RNA sequencing.

## Image analysis

Images were collected using a widefield Deltavision Elite microscope at ×60 magnification. All images were deconvoluted by Deltavision's default software, SoftWorx. Images of monolayers were collected as large stacks (40–70 images) with many slices 0.2 μm apart from each other. FIJI (*Schindelin et al., 2012*) was used to filter all images in a stack using the default 'rolling ball' algorithm before being average or max projected through the XY, XZ, and YZ planes by the XYZ projection plugin (*Omer et al., 2018*). Average projections were used for all protein signal while max projections were used for smFISH signals. Projections were brightness and contrast matched for all relevant samples. RGB profile lines were drawn and analyzed with RGB profiler (*Christophe Laummonerie, 2004*).

CAD neurites were also imaged as stacks (20–50 images) with many slices 0.2 μm apart from each other. Images were filtered using the default 'rolling ball' algorithm and max projected through the XY.

## *PRRE identification in 5′ UTRs*

Genes that were translationally down regulated upon mTOR inhibition (*Hsieh et al., 2012*) were assayed for pyrimidine-rich stretches (at least 9 pyrimidines in a 10 nucleotide window) across their longest annotated 5′ UTRs. The position of these PRTEs was defined as the most 5′′ start position of pyrimidine richness. If the position of the motif was 1, or the first nucleotide in the 5′ UTR, the PRTE could be more accurately defined as a 5′ TOP motif. Because this manuscript demonstrates that these pyrimidine-rich motifs can regulate localization in addition to translation, we refer to these motifs more broadly as pyrimidine-rich regulatory elements (PRREs).

## Neuronal fractionation sequencing computational strategy

Transcript abundances were calculated by salmon v0.11.1 (*Patro et al., 2017*) using Gencode genome annotations. Transcript abundances were then collapsed to gene abundances using txImport (*Soneson et al., 2015*). Localization ratios were calculated for each gene as the $\log_2$ of the ratio of neurite/soma

normalized counts for a gene produced by DESeq2 using the salmon/tximport calculated counts (*Love et al., 2014*). Genes were required to have a minimum of 10 counts in any sample for analysis.

## siRNA depletion of Larp proteins

Mouse-specific siRNAs were ordered from IDT for each Larp target. siRNAs were transfected into cells with Lipofectamine RNAiMAX (Thermo Fisher, 13778100). Briefly, 0.5 µL of 10 µM siRNA was combined with 50 µL Optimem media and 1.5 µL RNAiMAX reagent for roughly $1.0 \times 10^5$ cells in one well of a 24-well plate. CAD cells were transfected in the presence of doxycycline to induce expression of reporter constructs. Media was replaced 24 hr after transfection with full growth media supplemented with 2 µg/mL doxycycline. 24 hr following media change, cells were plated on PDL-coated coverslips (if imaging) and after attaching, media was changed again for serum-free media with 2 µg/mL doxycycline to induce neurite outgrowth for 48 hr. Negative control siRNAs from IDT were used to control for effects due to transfection. For all proteins tested, knockdown of expression was confirmed by qPCR normalized to HPRT. For Larp1, immunoblots confirming protein knockdown were performed with mouse monoclonal Larp1 antibody (Santa Cruz sc-515873).

## qPCR of endogenous transcripts

SYBR green (Biorad) mastermix was used to analyze RNA knockdown of Larp siRNAs and LARP1 rescue in C2bbe1 cells. RNA was collected from siRNA transfected CAD cells as detailed above. Total RNA was collected from C2bbe1 monolayers plated at high confluency in 12 well plates. For C2bbe1 cells, media containing 2 µg/mL doxycycline was changed every 3–4 days for 7 days of differentiation. Following differentiation of both cell types, media was removed from cells and replaced with RNA Lysis Buffer (Zymo Research, R1060-1). Cells were lysed at room temperature for 15 min on a rocker. RNA was collected using Quick RNA micro columns (Zymo Research, R1051). The provided on-column DNAse treatment was performed for 15 min at room temperature. 500 ng of purified RNA was used to synthesize cDNA using iScript Reverse Transcriptase Supermix (BioRad, 1708841). cDNA was combined with gene specific primers and iTaq Universal SYBR green master mix (BioRad, 1725122). Gene expression of siRNA targeted Larps was normalized to HPRT before being compared to the expression in negative control siRNA targeted samples.

## Immunoblotting

Expression of Larp1 siRNA treated and CRISPR/Cas9 KO cell lines were determined by immunoblot analysis. Lysates were collected by scraping cells into 100 µL RIPA buffer, incubated on ice for 15 min then sonicated for 30 s. Lysates were stored at –80 °C. Lysates were denatured with sample buffer (Invitrogen NP0008) and boiling at 100 °C for 5–10 min. Denatured lysates were separated by PAGE on 4%–12% Bis-Tris gradient gels along with Flash Protein Ladder (Gel Company FPL-008) using MOPS SDS NuPAGE Running Buffer (Thermo Fisher, NP0001) at 200 V for 1 hr. Gels were transferred to PVDF membranes using an iBlot2 dry transfer device (Thermo Fisher, IB24002). Total protein was assayed by incubation with Ponceau staining (Sigma-Aldrich P7170), which was later destained and imaged on a Sapphire biomolecular imager (Azure Biosystems). Blots were then blocked with agitation in 5% milk powder (Sigma) in PBST (PBS +0.01% Tween) for 1 hr at room temperature. Blots were washed three times, 5 min each in PBST at room temperature with agitation before being incubated in 1:100 LARP1 mouse monoclonal (Santa Cruz) primary antibody in 2% BSA (Research Products International A30075) in PBST overnight at 4 °C with agitation with 1:5000 beta actin mouse monoclonal antibody (Sigma A5441) in 2% milk powder in PBST for 2 hr at room temperature with agitation or with 1:10,000 H3 mouse monoclonal antibody (Abcam ab10799) in 2% milk powder in PBST for 2 hr at room temperature with agitation. Blots were washed three times, 5 min each at room temperature with agitation and then incubated with Anti-mouse IgG HRP-conjugated secondary antibody (Cell signaling 7076 S) in 2% milk powder in PBST and incubated at room temperature for 2 hr. Blots were washed again in PBST and visualized using WesternBright HRP Substrate Kit (Advansta, K-12043-D10) and Sapphire biomolecular imager (Azure Biosystems) set to collect chemiluminescent signal.

## Neuronal Fractionation of CAD cells for Taqman qPCR

Generally, neuronal fractionation experiments were performed as previously described (*Arora et al., 2021*). To fractionate CAD cells into soma and neurite fractions, CAD cells were plated on porous,

transwell membranes that had a pore size of 1.0 micron (Corning 353102). One million CAD cells were plated on each membrane. These membranes fit in one well of a six well plate, and three membranes were combined to comprise one single biological replicate. After allowing the cells to attach, the cells were induced to differentiate into a more neuronal state through the withdrawal of serum. Cells were then differentiated on the membranes for 48 hr.

To fractionate the cells, the membranes were washed one time with PBS. One mL of PBS was added to the top (soma) side of the membrane. The top side of the membrane was then scraped repeatedly and thoroughly with a cell scraper and the soma were removed into an ice-cold 15 mL falcon tube. After scraping, the membranes were removed from their plastic housing with a razor blade and soaked in 500 μL RNA lysis buffer (Zymo R1050) at room temperature for 15 min to make a neurite lysate. During this time, the 3 mL of soma suspension in PBS was spun down and resuspended in 300 μL PBS. Of this soma sample, 100 μL was then carried forward for RNA isolation using the Zymo QuickRNA MicroPrep kit (Zymo R1050). In parallel, RNA was also isolated from the 500 μL neurite lysate using the same kit. Typically, the neurite RNA yield from the six combined membranes was 100–300 ng. The efficiency of the fractionation was assayed by Western blotting using anti-beta-actin (Sigma A5441, 1:5000) and anti-Histone H3 (Abcam ab10799, 1:10000) antibodies. cDNA was created from 100 ng of RNA from each fraction using LunaScript RT supermix kit (NEB #E3010) following the manufacturer's instructions.

## Taqman qPCR of neurite fractions

All Firefly luciferase reporter constructs were driven by a bi-directional tet-On promoter that also induces Renilla luciferase expression. Renilla was never modified and acts as an internal control in these experiments. Expression of Firefly and Renilla luciferase transcripts was induced by incubating cells with 2 μg/mL doxycycline for 48 hr. qPCR was performed in BioRad thermal cyclers with taqman mastermix (Life Technologies 4444557) and probes specific to each luciferase but with different fluorophores (VIC and FAM). The relative expression of Firefly and Renilla luciferase was quantified in the same sample using Taqman qPCR then compared between neuronal fractions to calculate a $\log_2$ normalized localization ratio.

Endogenous mouse Rpl7 and Rps28 localization was also assayed by Taqman qPCR. Probes specific to these genes were designed with fluorophores FAM. A non-localized endogenous transcript, Tsc1 (VIC) was used as an internal control. The relative expression of these genes was quantified in the same sample to calculate $\log_2$ normalized localization ratios (LR). The endogenous Rpl7 and Rps28 LRs were compared to Firefly expression also normalized to Tsc1.

All qPCRs were performed with triplicate technical replicates averaged to produce a single biological replicate. qPCR experiments usually have three to four biological replicate samples included.

## Morphology analysis of CAD cells

CAD cells were siRNA treated with various targeting or negative control siRNAs as detailed above. Differentiation occurred in a single well of a 6-well plate. An EVOS imager (Thermo Fisher) at 20 X was utilized to take tiled brightfield images across the entire well. Images were analyzed in FIJI (*Schindelin et al., 2012*) using high contrast settings and NeuronJ (*Meijering, 2004*) was used to measure neurite lengths. The same images were used to count cells with and without neurites for qualitative morphological analyses.

## Published data sets used

*Moor et al., 2017*, Mouse LCM RNAseq (GSE95416: GSE95276; PRJNA377081).

*Dermit et al., 2020*, projection localization RNA seq (E-MTAB-8470, E-MTAB-9520, and E-MTAB-9636).

*Cassella and Ephrussi, 2022*, *Drosophila* Follicular Cell LCM RNAseq (E-MTAB-9127).

## Materials availability

All materials including cell lines and reagents are available upon request.

## Acknowledgements

We thank members of the Taliaferro lab for helpful discussions regarding experiments and analyses. We further thank Rytis Prekeris and Peter Dempsey for helpful discussions and advice regarding epithelial cell biology. Portions of the figures in this manuscript were created using Biorender. This work was funded by the National Institutes of Health (R35-GM133385 to JMT), the WM Keck Foundation (JMT) and the RNA Bioscience Initiative at the University of Colorado Anschutz Medical Campus (JMT, RG). This work was further supported by a Predoctoral Training Grant in Molecular Biology NIH-T32-GM008730 to RG.

---

## Additional information

### Funding

| Funder | Grant reference number | Author |
| --- | --- | --- |
| National Institutes of Health | R35-GM133385 | J Matthew Taliaferro |
| W. M. Keck Foundation | | J Matthew Taliaferro |
| National Institutes of Health | T32-GM008730 | Raeann Goering |

The funders had no role in study design, data collection and interpretation, or the decision to submit the work for publication.

### Author contributions
Raeann Goering, Conceptualization, Formal analysis, Investigation, Visualization, Methodology, Writing - original draft, Writing - review and editing; Ankita Arora, Resources, Investigation, Visualization; Megan C Pockalny, Investigation, Visualization; J Matthew Taliaferro, Conceptualization, Resources, Formal analysis, Funding acquisition, Investigation, Methodology, Writing - original draft, Project administration, Writing - review and editing

### Author ORCIDs
Raeann Goering ![ORCID] http://orcid.org/0000-0002-0351-335X
Ankita Arora ![ORCID] http://orcid.org/0000-0002-0279-8349
J Matthew Taliaferro ![ORCID] http://orcid.org/0000-0001-7580-1433

### Decision letter and Author response
Decision letter https://doi.org/10.7554/eLife.80040.sa1
Author response https://doi.org/10.7554/eLife.80040.sa2

---

## Additional files

### Supplementary files
• Supplementary file 1. Localization values in C2bbe1 monolayers. DESeq2 output of Cytoplasmic Bias (CB) and calculated FDR values (CB_FDR). Cytoplasmic bias is the $\log_2$ of a gene's cytoplasmic enriched abundance divided by the abundance in total RNA. Apical Bias (AB) and AB_FDR values are also included for all observed genes where Apical bias is the $\log_2$ of a gene's apical enriched abundance divided by its basal enriched abundance.

• Supplementary file 2. smFISH probe sequences. Sequences of individual probes used as equimolar pool to visualize endogenous RNAs in fixed cells. Note that the last 28 nt of each probe correspond to the sequence that hybridize to the Cy3-labeled secondary probe.

• MDAR checklist

### Data availability
All high-throughput RNA sequencing data as well as transcript quantifications have been deposited at the Gene Expression Omnibus under accession number GSE200004.

The following dataset was generated:

| Author(s) | Year | Dataset title | Dataset URL | Database and Identifier |
|---|---|---|---|---|
| Goering R, Arora A, Taliaferro JM | 2022 | RNA localization across the apicobasal axis of human epithelial cells | https://www.ncbi.nlm.nih.gov/geo/query/acc.cgi?acc=GSE200004 | NCBI Gene Expression Omnibus, GSE200004 |

The following previously published datasets were used:

| Author(s) | Year | Dataset title | Dataset URL | Database and Identifier |
|---|---|---|---|---|
| Moor AE, Golan M, Massasa EE, Lemze D, Weizman T, Shenhav R, Baydatch S, Mizrahi O, Winkler R, Golani O, Stern-Ginossar N, Itzkovitz S | 2017 | Global mRNA polarization regulates translation efficiency in the intestinal epithelium | https://www.ncbi.nlm.nih.gov/geo/query/acc.cgi?acc=GSE95416 | NCBI Gene Expression Omnibus, GSE95416 |
| Dermit M, Dodel M, Fcy L, Azman MS, Schwenzer H, Jones JL, Blagden SP, Ule J, Mardakheh FK | 2020 | 3' mRNA-seq of protrusions and cell bodies of BJ, PC-3M, RPE-1, U-87 and WM-266.4 cells | https://www.ebi.ac.uk/arrayexpress/experiments/E-MTAB-8470/ | ArrayExpress, 8470 |
| Cassella L, Ephrussi A | 2022 | *Drosophila* Follicular Cell LCM RNAseq | https://www.ebi.ac.uk/biostudies/arrayexpress/studies/E-MTAB-9127 | Array Express, 9127 |

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
