## [Editor Report]

In this valuable work, the authors used a combination of RNA-seq-based approaches and reporter mRNAs coupled to RNA imaging, to provide solid evidence that mRNAs with specific elements (TOP, GA, and NET1 3'UTR) localize asymmetrically across species and cell types and that this is likely mediated by conserved RBPs and via direct transport mechanisms preferentially involving kinesin motors. The work will be of interest to the cell biology community focused on mRNA localization.

---

## [Decision Letter]

**Decision letter after peer review:**

Thank you for submitting your article "RNA localization mechanisms transcend cell morphology" for consideration by *eLife*. Your article has been reviewed by 2 peer reviewers, and the evaluation has been overseen by a Reviewing Editor and James Manley as the Senior Editor. The following individual involved in the review of your submission has agreed to reveal their identity: Jeffrey Chao (Reviewer #1).

The reviewers have discussed their reviews with one another, and the Reviewing Editor has drafted this to help you prepare a revised submission. Both reviewers feel the manuscript is worthy of publication but they feel it would be stronger with some additional data, as described below.

Essential revisions:

1. There are several points related to the smFISH experiments that were brought up by both reviewers that need to be addressed that will require a modest amount of experiments as well as additional image analysis

2. There are several control experiments requested along with clarification of the 5' TOP sequences and reporters that were used in their experiments. This is important because the current results are inconsistent with the published literature as presented.

3. It will be important to verify the asymmetric localization of the endogenous ribosomal protein-encoding mRNAs characterized in this study. The reviewers could not find the pertinent data in the manuscript.

Further details are available in the full reviews below.

*Reviewer #1 (Recommendations for the authors):*

– Line 14: "results".

– Figure 1-S2: Please add marker lengths to the gel image to indicate the size of PCR products. How was integration validated? Sequencing of PCR-product?

– Line 63: The authors use a system that allows Dox-inducible expression of transgenes. They describe the integration of a Tet-on promoter that responds to Dox treatment, but not the transactivator necessary for induction

– Figure 1, which Halo ligands were used for imaging?

– Figure 2-S2: Please label marker bands. Is the second band for the apical fusion protein expected?

– Figure 2-S3 and 4: Are the scales for intensity (y-axis) identical between figures?

– Line 196:.… confirmed the "ability" of the apical-Halo…

*Reviewer #2 (Recommendations for the authors):*

1. For all the RNA imaging-based experiments you quantified a very limited number of cells (usually ~10 cells) and it is not clear if the quantifications are derived from independent experiments. You mentioned in the methods that you used FISH-quant for the quantifications, which has a very powerful batch mode for the quantifications of many images at the same time. In papers where smFISH is performed on neurons, usually between 30 and 100 cells are counted per condition. increasing cell counts would strongly strengthen your conclusions and statistical analysis.

2. For all experiments where RNA is quantified by smFISH or smiFISH please always report not only the ratios but also the absolute counts of mRNA transcripts per cell. You do so for some of the reporters and conditions, Figures 2, 3, and 6, but not for Figures 4 and 5. This information is important to account for the potential differences in gene expression of the different reporters.

3. The immunoblots verifying the downregulation of the RBPs shown in Figure 4 – —figure supplement 2 should be provided in the supplementary material, not just the ones for Larp1.

4. A control experiment where the smFISH for the endogenous mRNAs RPL7 or RPS28 should be provided in addition to the reporter constructs. These two mRNAs are the ones that you have extensively characterized throughout the paper. I believe that it would be important to show the bias in their localization both in epithelial and neuronal cell lines, even though single molecule detection may be difficult due to the high mRNA counts. If single mRNAs cannot be quantified, total integrated fluorescence intensity can be measured in different compartments to estimate the localization bias.

---

## [Author Response]

Essential revisions:1. There are several points related to the smFISH experiments that were brought up by both reviewers that need to be addressed that will require a modest amount of experiments as well as additional image analysis

We appreciate the reviewers’ concerns about how the analysis of smFISH images that we performed was inappropriate. In particular, we understand the concerns about how counting each individual smFISH punctum as a unique observation may artificially overpower the results.

To address this in the C2bbe1 epithelial system, we reanalyzed smFISH images using each of the following as a unique observation: the mean location of all smFISH puncta in one cell, the mean location of all puncta in a field of view, and the mean location of all puncta in one coverslip. Using each of these metrics, the results we observed were very similar (Author response image 1) with the obvious difference of reductions in statistical power. We therefore chose to reanalyze the existing images using the mean locations of puncta within individual cells as our metric of choice. Figures 2G, 3C, 3F, 4F, 4G, 5B, 5D, 6D, 7C, 7F, and several supplemental figures have been updated to reflect this change.

**Author response image 1. sa2fig1:** C2bbe1 monolayer smFISH spot position analysis. RNA localization across the apicobasal axis is measured by smFISH spot position in the Z axis. This can be plotted for each spot, where thousands of spots over-power the statistics. Spot position can be averaged per cell as outlined manually within the FISH-quant software. This reduces sample size and allows for more accurate statistical analysis. When spot position is averaged per field of view, sample size further decreases, statistics are less powered but the localization trends are still robust. Finally, we can average spot position per coverslip, which represents biological replicates. We lose almost all statistical power as sample size is limited to 3 coverslips. Despite this, the localization trends are still recognizable..

For the CAD neuronal smFISH images, those analyses were already done on a per-cell level as the metric presented is neurite puncta / soma puncta within one individual cell. Still, the reviewers’ comments about needing to image more cells for these experiments are worthwhile. Rather than imaging more cells, though, we turned to the completely orthogonal technique of subcellular fractionation into soma and neurite fractions followed by RT-qPCR. This technique profiles the population average of approximately 3 million cells. The results from fractionation/RT-qPCR experiments agreed very well with the original smFISH results (compare Figure 3M and Figure 3 —figure supplement 13). We hope that the use of two orthogonal techniques to quantify neurite RNA localization lends increased confidence to the results.

2. There are several control experiments requested along with clarification of the 5' TOP sequences and reporters that were used in their experiments. This is important because the current results are inconsistent with the published literature as presented.

The reviewers’ point about whether pyrimidine-rich motifs need to be directly adjacent to the 5’ cap in order to regulate localization is a very good one. There is conflicting literature about whether LARP proteins, and in particular LARP1, require such an arrangement. Further, the nomenclature in the literature is confusing as some say that in order for pyrimidine-rich motifs to be called TOP motifs they must be immediately next to the cap while others are more lenient with their position with the 5’ UTR.

Published CAGE data indicates that a substantial fraction of endogenous RPL7 and RPS28 transcripts begin with a purine (see **reviewer figure 2**). Yet, endogenous RPL7 and RPS28 transcripts are clearly basally/neurite enriched (figure 3C and 3K).

A deeper analysis of our reporter transcripts led us to find that the pyrimidine-rich elements within our reporter RNAs were located approximately 90 nt into the body of the 5’ UTR. Yet, these reporter transcripts are also basally/neurite localized. We redesigned our reporters such that the pyrimidine-rich motif was immediately adjacent to the 5’ cap and found that their localization was very similar to those with internal pyrimidine-rich motifs (see **reviewer figure 3**). These results suggest that a pyrimidine-rich element need not be immediately adjacent to the 5’ cap to regulate RNA localization.

To generalize these results, we began with pyrimidine-rich 5’ UTR elements that had previously been found to regulate translation in an mTOR-dependent manner (Hsieh et al. 2012). We found that RNAs containing these elements were similarly basally/neurite localized as TOP-containing RP mRNAs (figures 3A, 3I). To assess any relationship between the location of pyrimidine-rich elements within 5’ UTRs and their localization regulatory ability, we compared the localization of transcripts with pyrimidine-rich elements at different 5’ UTR positions. We found no such relationship, and motifs in the body of the 5’ UTR seemed just as able to regulate RNA localization as those immediately adjacent to the 5’ cap. This data is presented in figures 3B and 3J.

From this, we conclude that localization-regulating pyrimidine-rich elements can be located anywhere within 5’ UTRs. We have therefore begun calling these elements “pyrimidine-rich regulatory elements” (PRREs) to differentiate them from 5’ TOP motifs since there is literature requiring 5’ TOP motifs to be cap-adjacent.

3. It will be important to verify the asymmetric localization of the endogenous ribosomal protein-encoding mRNAs characterized in this study. The reviewers could not find the pertinent data in the manuscript.

To address this point, we have performed smFISH on the two endogenous RP-mRNAs we used as the basis for our reporters, RPL7 and RPS28. In epithelial cells, we found their endogenous transcripts to be extremely basally enriched. This data can be found in figure 3C and figure 3 supplement 3. For neuronal cells, we again turned to the orthogonal subcellular fractionation / RT-qPCR experiment. Using our nonlocalized firefly reporter as a control, we found that endogenous mouse Rpl7 and Rps28 transcripts were approximately 4-fold neurite enriched. This data is included as figure 3K.

Further details are available in the full reviews below.Reviewer #1 (Recommendations for the authors):– Line 14: "results".

Thanks for catching this. This mistake has been corrected.

– Figure 1-S2: Please add marker lengths to the gel image to indicate the size of PCR products. How was integration validated? Sequencing of PCR-product?

Marks indicating the lengths of the observed products have been added. PCR products were not sequenced, but PCR reactions were designed that flanked both newly created junctions such that products would only be produced if the insert had been integrated. Both of these products were found in the integration line but not in a control line in which the insertion had not been performed (Figure 1 supplement 2). This has now been more thoroughly explained in the methods.

– Line 63: The authors use a system that allows Dox-inducible expression of transgenes. They describe the integration of a Tet-on promoter that responds to Dox treatment, but not the transactivator necessary for induction

The plasmid that contains the dox-activatable transgene also contains a constitutively driven transactivator. This has now been more thoroughly explained in the methods under the header “Cre recombinase-mediated cassette switching”.

– Figure 1, which Halo ligands were used for imaging?

For the in cell imaging, JF549 was used. For the in-gel protein staining, JF646 was used. This information, including catalog numbers, has been added to the methods under the heading “Halo imaging in fixed cells”.

– Figure 2-S2: Please label marker bands. Is the second band for the apical fusion protein expected?

Markers indicating protein sizes of ladder bands have been added. The reviewer is correct that the apical fusion protein does display two bands on this gel. Based on the design of our transgene, we did not expect this. It is possible that some cleavage or other protein processing is happening with this protein. Regardless, figure 2A demonstrates that almost all apical fusion protein, no matter the potential cleavage product, is apically localized, indicating that it can still serve as a useful reagent for proximity labeling.

– Figure 2-S3 and 4: Are the scales for intensity (y-axis) identical between figures?

Yes, these intensity scales are identical.

– Line 196:.… confirmed the "ability" of the apical-Halo…

This mistake has been corrected.

Reviewer #2 (Recommendations for the authors):1. For all the RNA imaging-based experiments you quantified a very limited number of cells (usually ~10 cells) and it is not clear if the quantifications are derived from independent experiments. You mentioned in the methods that you used FISH-quant for the quantifications, which has a very powerful batch mode for the quantifications of many images at the same time. In papers where smFISH is performed on neurons, usually between 30 and 100 cells are counted per condition. increasing cell counts would strongly strengthen your conclusions and statistical analysis.

We appreciate this comment and understand the necessity of high cell numbers for confidence in results. For the epithelial cell images, we have reorganized the analysis such that the reported metric is now mean transcript position per cell. In these images, we are routinely assaying dozens to hundreds of cells per condition.

For the neuronal images, it is more difficult to image many cells as their extended morphology makes it difficult to capture an entire cell in one field of view. Given that we would have to perform additional imaging on dozens of individual cell lines, we instead turned to an orthogonal technique to bolster the neuronal smFISH results. Using porous transwell membranes, we mechanically fractionated these cells into soma and neurite fractions, collected RNA from each fraction, and then quantified endogenous and reporter transcript abundances in each fraction using RT-qPCR. This technique reports the population average from approximately 3 million cells across at least three biological replicates.

In all cases, results from the subcellular fractionation agreed quite well with the smFISH results (compare figure 3M with figure 3 —figure supplement 13). We hope that the use of an orthogonal technique to quantify neuronal RNA localization lends sufficient additional confidence to the results.

2. For all experiments where RNA is quantified by smFISH or smiFISH please always report not only the ratios but also the absolute counts of mRNA transcripts per cell. You do so for some of the reporters and conditions, Figures 2, 3, and 6, but not for Figures 4 and 5. This information is important to account for the potential differences in gene expression of the different reporters.

This is an important point. Transcript counts per cell have now been included for the experiments in figure 3 (supplements 4, 5, 6, 8, and 12), figure 4 (supplements 9 and 11), figure 5 (supplements 1, 3, 6, and 8), figure 6 (supplements 1 and 2), and figure 7 (supplements 2, 4, 5, and 6).

3. The immunoblots verifying the downregulation of the RBPs shown in Figure 4 – —figure supplement 2 should be provided in the supplementary material, not just the ones for Larp1.

We agree that this would be helpful, but we do not have robust antibodies for each of the LARP family members. For each tested LARP, we did verify knockdown using RT-qPCR and found the knockdowns to be strong (>80%, figure 4 —figure supplement 1). We understand that knockdown at the RNA level does not necessarily equal knockdown at the protein level, but without well-validated antibodies, this is the best we can do.

4. A control experiment where the smFISH for the endogenous mRNAs RPL7 or RPS28 should be provided in addition to the reporter constructs. These two mRNAs are the ones that you have extensively characterized throughout the paper. I believe that it would be important to show the bias in their localization both in epithelial and neuronal cell lines, even though single molecule detection may be difficult due to the high mRNA counts. If single mRNAs cannot be quantified, total integrated fluorescence intensity can be measured in different compartments to estimate the localization bias.

This is an important point. In epithelial cells, we visualized endogenous RPL7 and RPS28 transcripts using smFISH. Both transcripts were extremely basally localized compared to an unlocalized firefly luciferase control transcript. This data has now been included as figure 3C and figure 3 supplement 3.

For looking at endogenous mouse Rpl7 and Rps28 transcripts in the neuronal system, we turned again to our subcellular fractionation into soma and neurite fractions followed by RT-qPCR. This technique is orthogonal to the smFISH approach and quantifies the population average of approximately one million cells. Again, endogenous Rpl7 and Rps28 were neurite-enriched by this method. This data has now been included as figure 3K.